# Pyro-catalysis for tooth whitening via oral temperature fluctuation

Yang Wang[1], Shuhao Wang[1], Yanze Meng[2], Zhen Liu[1], Dijie Li[1], Yunyang Bai[3], Guoliang Yuan [1], Yaojin Wang [1] ✉, Xuehui Zhang[2,4] ✉, Xiaoguang Li [5] & Xuliang Deng [3,4] ✉

Tooth whitening has recently become one of the most popular aesthetic dentistry procedures. Beyond classic hydrogen peroxide-based whitening agents, photo-catalysts and piezo-catalysts have been demonstrated for non-destructive on-demand tooth whitening. However, their usage has been challenged due to the relatively limited physical stimuli of light irradiation and ultrasonic mechanical vibration. To address this challenge, we report here a non-destructive and convenient tooth whitening strategy based on the pyro-catalysis effect, realized via ubiquitous oral motion-induced temperature fluctuations. Degradation of organic dyes via pyro-catalysis is performed under cooling/heating cycling to simulate natural temperature fluctuations associated with intake and speech. Teeth stained by habitual beverages and flavorings can be whitened by the pyroelectric particles-embedded hydrogel under a small surrounding temperature fluctuation. Furthermore, the pyro-catalysis-based tooth whitening procedure exhibits a therapeutic biosafety and sustainability. In view of the exemplary demonstration, the most prevalent oral temperature fluctuation will enable the pyro-catalysis-based tooth whitening strategy to have tremendous potential for practical applications.

Tooth discoloration is becoming an increasingly common aesthetic problem because it jeopardizes people's appearance and self-confidence[1]. Moreover, tooth discoloration readily results from external chromogens typically consumed in various food and beverages[2,3]. As a result, tooth whitening has developed as one of the most popular aesthetic dentistry procedures[4,5]. Clinical tooth bleaching practices are generally based on high-concentration active carbamide peroxide and hydrogen peroxide[6], which can generate superoxide $\cdot O_2^-$ and $\cdot OH$ radicals during their decomposition. These radicals further break the conjugated double bonds of organic pigment molecules and then degrade staining macromolecules into light compounds by oxidation

reactions[7]. Despite their highly efficient whitening effect, tooth bleaching products can cause serious side effects associated with robust and high-concentration oxidant radicals, such as gingival irritation[8], mineral loss, and tooth hypersensitivity[9–11].

As the generation of reactive oxygen species (ROS) is the backbone of tooth whitening, photo-catalysis and piezo-catalysis have been proposed as promising alternative strategies to produce oxidative radicals[2,12]. It has been demonstrated that agents embedded with $TiO_2$ particles can be used for effective and non-destructive tooth whitening under blue-light activation[12]. In particular, an emergent non-destructive and harmless tooth whitening strategy wherein typical

[1]School of Materials Science and Engineering, Nanjing University of Science and Technology, Nanjing 210094 Jiangsu, China. [2]Department of Dental Materials & Dental Medical Devices Testing Center, Peking University School and Hospital of Stomatology, Beijing 100081, China. [3]Department of Geriatric Dentistry, Peking University School and Hospital of Stomatology, Beijing 100081, China. [4]National Engineering Research Center of Oral Biomaterials and Digital Medical Devices, NMPA Key Laboratory for Dental Materials, Beijing Laboratory of Biomedical Materials & Beijing Key Laboratory of Digital Stomatology, Peking University School and Hospital of Stomatology, Beijing 100081, China. [5]Hefei National Laboratory for Physical Sciences at the Microscale, Department of Physics, and CAS Key Laboratory of Strongly-Coupled Quantum Matter Physics University of Science and Technology of China, Hefei 230026, China. ✉e-mail: yjwang@njust.edu.cn; zhangxuehui@bjmu.edu.cn; kqdengxuliang@bjmu.edu.cn

toothpaste abrasives were replaced with piezoelectric BaTiO₃ (BTO) particles has shown that routine daily toothbrushing itself without extra time-consuming and additional equipment can generate ROS necessary for effective tooth whitening[2]. However, such photo-catalysis and piezo-catalysis-based tooth whitening procedures have been challenged due to either photo-allergic reactions, or the unavailable intrinsic physical stimuli of light irradiation and ultrasonic mechanical vibration (Supplementary Table 1)[13,14]. Therefore, an effective, non-destructive and harmless tooth whitening procedure that requires no additional external energy or specialized equipment is in demand.

Pyroelectric materials with spontaneous electric polarization have the ability to generate an electric charge in response to temperature fluctuations[15]. Thus, the most representative application of pyroelectric materials is to detect real-time heat signals[16–18]. Additionally, pyro-electrics are often used in thermal energy harvesting applications for self-powered devices under surroundings with notable temperature changes[19–21]. Since electric charges can be generated by temperature fluctuation, pyroelectric materials have also been employed as catalysts (i.e., termed as pyro-catalysis) for hydrogen evolution[22], carbon dioxide reduction[23], and sewage disposal[24]. As an environmental-friendly lead-free pyroelectric material, BaTiO₃ (BTO) has attracted much attention in the field of pyro-catalysis, such as dye degradation by light-induced temperature change and waste heat[25–27], and waste-water treatment with assistance of metal nanoparticles[28]. The basic working principle of pyro-catalysis is similar to that of photo-catalysis and piezo-catalysis, i.e., generation of ROS in response to the external stimuli, but the stimuli of pyro-catalysts are temperature fluctuations, rather than the light or mechanical vibration for photo- and piezo-catalysts. Although the pyro-catalysis and photo-catalysis exhibit the same electrochemical-redox process[29–31], the pyro-catalysis has drawn less technological attention relative to photo-catalysis due to its power-based temperature fluctuation stimuli.

Temperature fluctuations are the most prevalent stimuli in our oral environment, when the mouth is open and close during speaking, drinking, and mouth breathing during exercise. (Fig. 1a). In view of these prevalent stimuli, we herein propose a non-destructive and safe tooth whitening strategy based on the pyro-catalysis effect, which is easily realized by braces containing pyroelectric particles embedded in a hydrogel. Here we demonstrate the degradation of organic dyes via pyro-catalysis performed under cooling/heating cycles to simulate habitual intake- and speech-induced temperature fluctuations. Teeth stained by habitual consumption of beverages and flavorings can be notably whitened by the pyroelectric particles-embedded hydrogel under small temperature fluctuations corresponding to natural oral temperature cycles. Furthermore, the pyro-catalysis-based tooth whitening system exhibits remarkably therapeutic biosafety and sustainability.

## Results

### Concept and feasibility of pyro-catalytic tooth whitening
The concept of pyro-catalysis for tooth whitening is generally to harvest ubiquitous oral motion-induced temperature fluctuation without additional equipment or power sources. As shown in Fig. 1b, oral temperature will change with typical daily activity. For example, when hot food (such as hot water) is ingested, oral temperature can rise from ~36 to ~48 °C. Conversely oral temperature will drop to ~21 °C when consuming cold food (such as ice cream). As is demonstrated in this work, this readily available physical stimulus can be used to excite the properties of pyroelectric materials and enables the application of pyro-catalysts for tooth whitening.

The screening charge on the surface of the pyroelectric material is released when a temperature fluctuation occurs. Exploitation of these temperature changed-induced surface charges as catalysts is called the pyro-catalytic effect or pyro-catalysis[32]. The typical pyro-catalysis process illustrated in Fig. 1c–e. When the pyroelectric material has been poled, the internal ferroelectric domains are arranged in an ordinal state, developing a heterogeneous charge (screening charge) on the surface of the material[33,34]. It has been reported that the intensity of the spontaneous polarization of pyroelectric materials decreases as temperature increases[35]. When the pyroelectric material is heated, the original orderly arrangement of ferroelectric domains will be somewhat randomized, resulting in a decrease in polarization. The change in internal polarization will decrease the required balancing surface charge and will result in the release of excess shielding charges on the surface. The released charge will combine with water molecules to form radicals (•OH or •O₂⁻) with strong redox properties[36]. Similarly, when a pyroelectric material is cooled, the decrease in temperature causes an increase in spontaneous polarization. As the thermodynamic equilibrium is broken once again, the surface of the material absorbs charge from the environment to make itself electrically neutral. When carried out in an aqueous environment, the remaining charge reacts with water molecules to form radicals[37].

From the perspective of a practical application, we have designed a dental brace that incorporates a pyroelectric material (Fig. 1f) that uses the pyro-catalytic to whiten teeth. These retainers can achieve pyro-catalysis through temperature fluctuations in the mouth induced by daily oral activities (e.g., drinking, breathing, talking, exercising, etc.), without using any other assistant equipment. The generated constant stream of active radicals will attack and degrade the stains on the tooth surface. Degradation of the colored macromolecular groups into small, colorless molecules results in a tooth whitening effect (Fig. 1g–i). As these dental braces require no active energy supply (photo stimulation, ultrasonic agitation, etc.), the act of whitening the teeth is essentially passive. Furthermore, typical temperature fluctuations in the mouth are well below the Curie temperature of the pyroelectric materials, resulting in extremely long lifetime of use.

### Synthesis and structural characterization of BaTiO₃ nanowires
As a demonstration, classical ferroelectric tetragonal BTO nanowires were synthesized by the hydrothermal method (see methods section) using H₂TiO₃ nanowires as the template crystals. Figure 2a shows the X-ray diffraction (XRD) pattern of the as-prepared BTO nanowires. All diffraction peaks can be well indexed to the perovskite structure and no impurity phases are detected. As the spontaneous polarization in BTO is due to the asymmetric distortion of the cubic perovskite structure, and because the diffraction technique has difficulty distinguishing the tetragonal phase from cubic symmetry (due to the broadened peak profile for samples with small crystallite size), Raman spectroscopic analysis was performed to investigate local distortions of the lattice. As shown in Fig. 2b, the distinct bands observed at 186, 249, 306, 515, and 715 cm⁻¹ in Raman scattering profiles can be assigned to the splitting of degenerated $3F_{1u} + F_{2u}$ modes of the polar crystal BTO (P4mm). In particular, the peak at 306 cm⁻¹ is usually associated with the asymmetric vibration of the [TiO₆] octahedra, thus the distinct phonon mode at 306 cm⁻¹ was regarded as the indicator of tetragonal distortion of the perovskite structure. The distortion from the ideal cubic perovskite structure allows the pyroelectric effect in the BTO nanowires.

Scanning electron microscopy (SEM) was used to characterize the morphology of the material (Fig. 2c). SEM results reveal a large quantity of nanowire materials with uniform geometrical morphology. The nanowires are ~100 nm in diameter, with 95% having lengths of ~5 μm, as shown in the TEM image of a typical BTO nanowire (Fig. 2d). The resulting aspect ratio (as calculated by length divided by diameter) of the nanowires is more than 50 for the majority of our samples.

The crystallographic structure of the BTO nanowires is further confirmed by high-resolution transmission electron microscopy (TEM). The high-resolution TEM image in Fig. 2e indicates a single crystalline structure with lattice fringes of (001). The interplanar

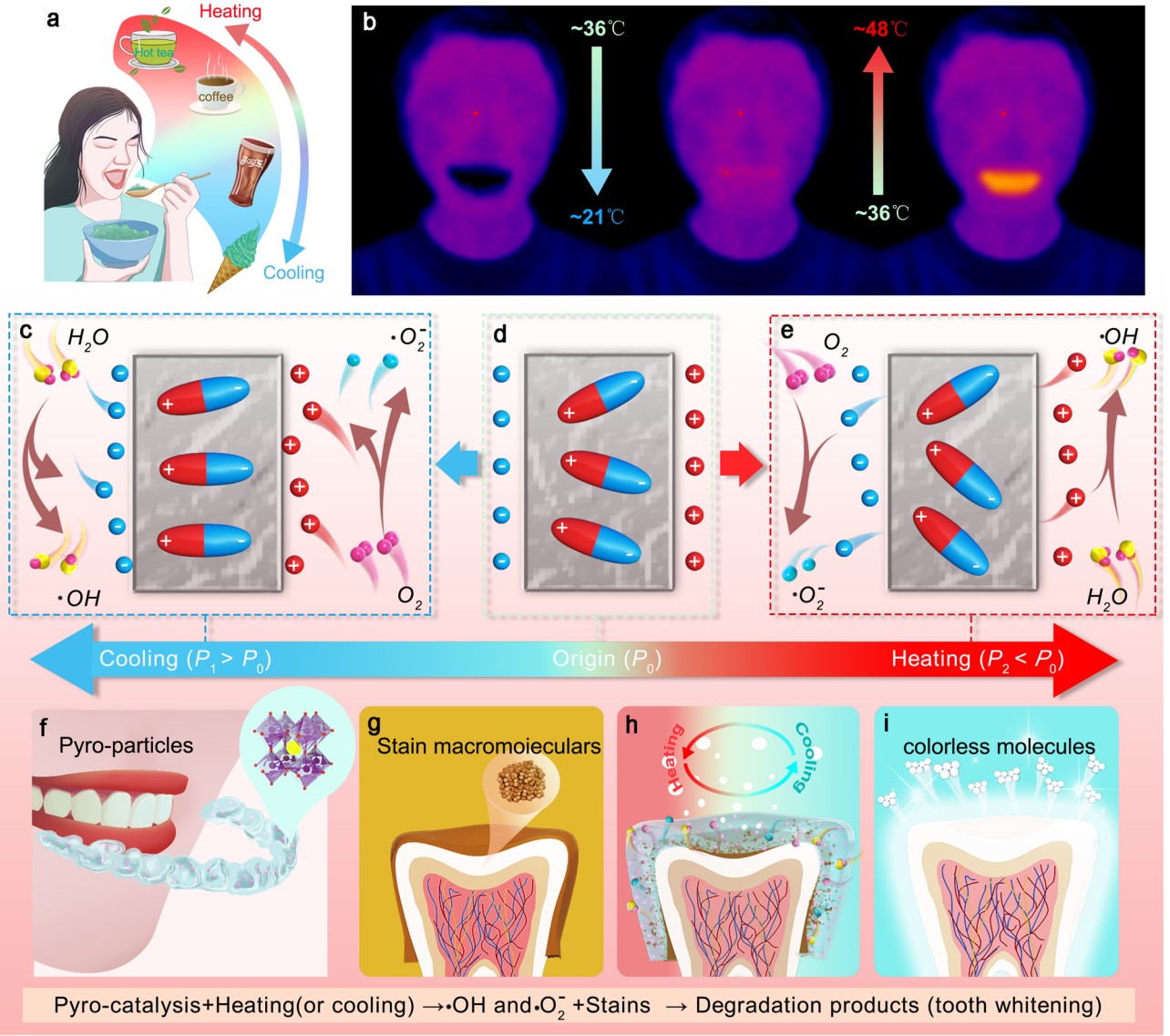

**Fig. 1 | Pyro-catalysis for tooth whitening. a** Consumption of different temperature beverages results in changes in oral temperature. **b** Infrared imaging of the change in oral temperature after drinking cold and hot water. **c–e** Schematic diagram of pyro-catalysis principle; heating or cooling leads to an alteration of the polarization strength in pyroelectric materials, which further causes the absorption and release of screening charges and the generation of reactive radicals in water. **f** The proposed pyro-catalysis effect-based tooth whitening method wherein pyroelectric particles are combined with light-curing hydrogel to generate reactive oxygen species via pyro-catalysis to bleach tooth stains. **g-i** Stained tooth can be whitened by wearing braces containing pyroelectric particles that use the pyro-catalytic properties activated by changes in oral temperature brought about by daily diet to degrade stains on the surface of the tooth.

spacing of 3.983 Å obtained by the selected area electron diffraction pattern (Fig. 2f) is consistent with the interplanar distance of (001) planes of BTO. The lattice parameter measurement confirms the tetragonal symmetry of the perovskite structure. Moreover, the orientation of (001) lattice fringes clearly shows the extension of BTO nanowires along the polar axis [001]. Furthermore, the STEM and energy dispersive X-ray spectroscopy elemental mappings (Fig. 2g–j) show, all the involved elements (Ba, Ti, and O) are homogeneously distributed and well-matched with each other, resembling the original STEM morphology.

The ferroelectricity of BTO nanowires was characterized by piezoelectric force microscopy (PFM). The diameter of the nanowires from Fig. 2k is on the order of ~90 nm. The piezoresponse magnitude can be estimated from the amplitude image of Fig. 2l. The phase image is shown in Fig. 2m, where bright and dark regions in the nanowires are corresponding to the domains oriented upwards and backwards directions, indicating the robust ferroelectricity of BTO nanowires.

The local piezoelectric hysteresis loop of a BTO nanowire is shown in Fig. 2n. The phase angle shows a full 180° change under a 5 V DC bias field. The phase switching, coupled with the butterfly shape of the amplitude loop implies the existence of well-defined polarizations along the vertical direction of the nanowires. And the finite element simulations of pyroelectric potential for different morphological nanomaterials showed that BTO nanowires exhibited outstanding pyroelectric potential compared to other nanostructures due to the spontaneous polarization along the length orientation. (Supplementary Fig. 1).

**Indigo Carmine degradation based on pyro-catalysis**
To characterize the pyro-catalytic properties of BTO nanowires and to evaluate the potential use as a tooth whitening agent, the common food additive Indigo Carmine was selected as the target contaminant for degradation experiments. Briefly, the degradation of Indigo Carmine solution was investigated using BTO nanowire turbid liquid with

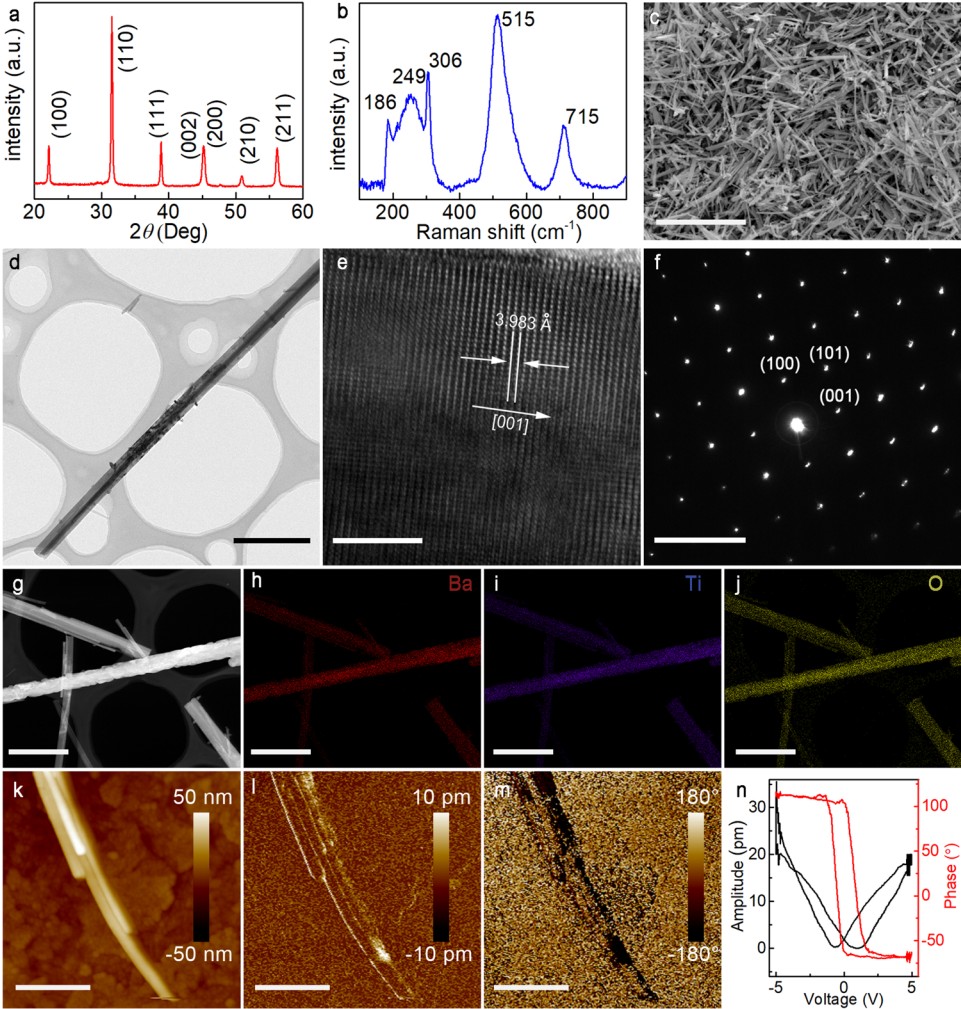

**Fig. 2 | Microstructural and morphology characterization. a** X-ray diffraction pattern of the BTO nanowires. **b** Room-temperature Raman spectra of the hydrothermal BTO nanowires. **c** Scanning electron microscope image of BTO nanowires, **d** Transmission electron microscope, **e** high-resolution transmission electron microscope images and **f** selected area electron diffraction patterns of the BTO nanowires, and **g–i** corresponding EDX element mapping of Ba (red), Ti (blue), and O (yellow) in BTO nanowires. PFM results of BTO nanowires **k** topography image; **l** vertical amplitude image; **m** vertical phase image and **n** piezoelectric hysteresis loop. Scale bar: **c** is 10 µm, d is 1 µm, **e** is 5 nm, **f** is 5 1/ nm, **g**, **h**, **i** and **j** are 1 µm, **k**, **l** and **m** are 500 nm. The experiments in **c–n** were repeated independently for three times with similar results. Source data are provided as a Source Data file.

a concentration of 1 mg mL⁻¹ under different temperature fluctuations. To simulate the temperature change in the human oral cavity more realistically, we chose 36 °C as the initial temperature and the pyro-catalysis experiment was carried out at different temperature ranges ($\Delta T$ = −10, −5, +5, +10, +15, +20 °C)[38] (Supplementary Fig. 2). The UV-Vis absorption spectra of an Indigo Carmine solution exposed to the BTO turbid liquid at various temperature fluctuations is presented in Fig. 3a–f. The maximum absorption peak of Indigo Carmine around 611 nm shows a notable decrease with increased number of thermal cycles, and the degradation rate can be higher than 98%. Impressively, at temperature fluctuation of $\Delta T$ = 20 °C, only three thermal cycles are required to degrade the Indigo Carmine (Supplementary Movie 1). In contrast, the degradation of Indigo Carmine was minimal when BTO nanowires were used as catalyst or when BTO nanowires were not added (Supplementary Fig. 3). The concentration of H₂O₂ monitored by liquid chromatography shows no change during the pyro-catalysis process (Supplementary Fig. 4), while the DPD-POD experimental results indicate the generation of 300 µM of H₂O₂ in the reaction system after 30 hot and cold cycles at a temperature difference of +20 °C (Supplementary Fig. 5). Anomalous to photo-catalysis, it can be inferred that H₂O₂ is as an intermediate product and remains an equilibrium state between consumption and generation during the

pyro-catalysis process. The charges released from the pyro-catalytic BTO nanowires was first combined with the addition of H₂O₂, and subsequently react further into reactive radicals in a Fenton-like reaction[39,40]. The addition of small amount of H₂O₂ can accelerate the pyro-catalytic process and improve the efficiency. These comparative experiments unambiguously verify that the degradation of Indigo Carmine due to the catalysis effect is strongly associated with the pyroelectricity of the nanowires.

To further verify the pyro-catalysis effect itself as a tooth whitening process, a duplicate experiment was carried out using [001]-oriented Pb(Mg$_{1/3}$Nb$_{2/3}$)-PbTiO₃ (PMN-PT) single crystal, a material with well-documented pyroelectric properties[41]. Bulk plate PMN-PT specimens were broken into pieces by ball-milling. After thermal cycling over the temperature ranges of interest, more than 95% of the Indigo Carmine was degraded when PMN-PT was used as the pyro-catalysis agent. Catalytic rate constant calculations show that the catalytic efficiency of a single heating/cooling cycle was improved as the temperature fluctuation range increases, with the same results observed when BTO nanowires were used as the catalyst (Supplementary Fig. 6).

It is worth noting that the degradation rate of Indigo Carmine solutions with the same concentration of pyro-catalyst is dependent

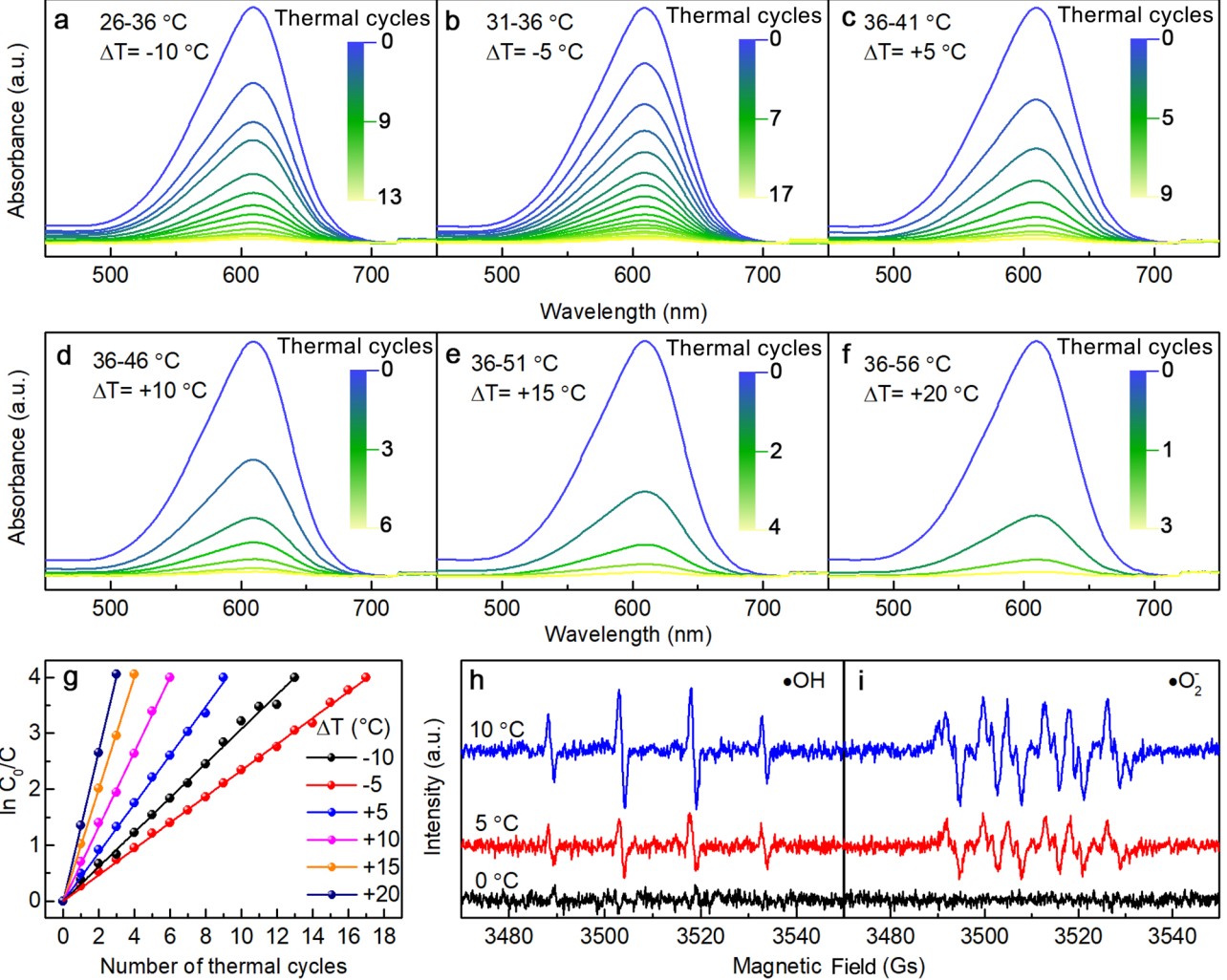

**Fig. 3 | Degradation properties of pyro-catalysis.** UV-Vis absorption spectra of Indigo Carmine solutions with respect to temperature fluctuations **a–f** $\Delta T$ = −10, −5, +5, +10, +15, +20 °C. **g** the pseudo-first-order reaction kinetics of different temperature fluctuations. Electron paramagnetic resonance spectra (EPR) of radical **h** •OH and **i** •$O_2^-$ created by pyro-catalysis over different temperature range. Source data are provided as a Source Data file.

on the absolute temperature, with the rate significantly faster at higher absolute temperature. For example, 13 thermal cycles were required for a 95% degradation of Indigo Carmine at 26–36 °C ($\Delta T$ = −10 °C), while only 6 thermal cycles were required for the same degradation of Indigo Carmine at 36–46 °C ($\Delta T$ = 10 °C) (Fig. 2a, d).

The pyroelectric effect relates the polarization ($P$), the temperature change ($\Delta T$), and the change in the surface charge of a pyroelectric material. The surface charge, $\Delta Q$ can be characterized by Eq. (1) when the temperature change ($\Delta T$) is known.

$$\Delta Q = p \cdot A \cdot \Delta T \qquad (1)$$

Where $p$ is the pyroelectric coefficient, and $A$ is the area of the surface from which the charge is released. From this equation we can obtain that for the same pyroelectric material, the charges released from the surface should be equal when experiencing the same temperature change. From this perspective, the number of thermal cycles required to degrade the Indigo Carmine solution at 26–36 °C and 36–46 °C should be the same which is different from the observed phenomenon. In fact, in the pyro-catalysis experiments, the entire process was carried out in aqueous solution, and instead of simply raising and lowering the temperature of the pyroelectric material, the entire reaction system underwent thermal cycling together. Temperature changes in a reaction system can drastically affect reaction kinetics,

resulting in accelerated degradation when the absolute system temperature is increased.

Although reaction kinetics are proportional to system absolute thermal energy, a comparison of experiments at 26–36 °C ($\Delta T$ = −10 °C) with those at 31–36 °C ($\Delta T$ = −5 °C) shows that the degradation rate is faster at 26–36 °C due to the greater temperature fluctuation experienced despite an overall decrease in average temperature. And it can be clearly seen from Fig. 3g that the number of cycles required for the degradation of Indigo Carmine decreases with increasing temperature fluctuation in the same temperature trend (heating or cooling). Therefore, we can conclude that the phenomenon observed in this experiment is still consistent with the theory, and the increased temperature variation range can induce more reactive radicals at the same temperature variation trend.

As natural oral temperature fluctuation rates vary (Supplementary Movie 2), the pyro-catalysis degradation experiment was carried out using different heating rates under the same cooling rate of 1 °C m$^{-1}$ (Supplementary Fig. 2b). The slower heating rate leads to a more efficient pyro-catalysis (i.e., $k_{Slow}$ = 0.467, $k_{Mid}$ = 0.303 and $k_{Fast}$ = 0.199, Supplementary Fig. 7), which can be attributed to a trade-off between positive and negative charge recombination and the ROS generation. The total charges induced by the pyroelectricity will be identical under the same temperature fluctuation (Eq. 1), while they may recombine to each other (Eq. 2), rather than release to the liquid to produce ROS

(Eqs. 3 and 4) at a rapid heating rate. In order to further discuss the charge recombination and release efficiency on cooling or heating rate, we have additionally performed pyro-catalysis experiment using different catalyst concentrations under the same temperature fluctuation (i.e., $\Delta T = +10\,°C$) and temperature rise rate (Supplementary Fig. 8). The pyro-catalytic efficiency increases and then decreases with the increase of catalyst concentration, indicating that charge release efficiency higher than a center degree will result in the recombination of charges with each other instead of reacting with water molecules to form radicals[42,43].

$$\text{Pyro} - \text{catalyst} + \text{Thermal cycles} \rightarrow \text{Pyro} - \text{catalyst} + (q^- + q^+) \quad (2)$$

$$q^- + O_2 \rightarrow \cdot O_2^- \quad (3)$$

$$q^+ + H_2O \rightarrow \cdot OH \quad (4)$$

$$\cdot OH \text{ or } \cdot O_2^- + \text{organic dye} \rightarrow \text{degradation products} \quad (5)$$

Typically, the pyro-catalyzed degradation of organic dyes is described as a series of chemical reactions (Eqs. 2–5). As illustrated in Fig. 1c, when a pyroelectric material is poled by an applied electric field, screening charges accumulate on its surface to balance the bound charges of electric polarization. The electric polarization will decrease with temperature increase, leading to a change in the bound and screening charge, and the excess screening charge will release into the aqueous solution to combine with water molecules to form radicals. Thermodynamically, the potential for generating •OH and •$O_2^-$ needs to be at least ~1.7 V and 1.9 V, respectively[44,45]. The pyroelectric potential ($\varnothing_{pyro}$) induced by temperature fluctuations ($\Delta T$) can be governed by Eq. (6).

$$\varnothing_{pyro} = \frac{p \cdot l \cdot \Delta T}{\varepsilon_0 \cdot \varepsilon_r} \quad (6)$$

where $p$ is the pyroelectric coefficient, $l$ is the length of nanowires, $\varepsilon_0$ is the permittivity of vacuum, $\varepsilon_r$ is the permittivity of BTO. The reported $p$ and $\varepsilon_r$ of BTO nanowires are ~210 μC m$^{-2}$ K$^{-1}$ and ~100[19,46]. It can be calculated that the required nanowire length is at least 1.6 μm when the temperature fluctuation $\Delta T = 5\,°C$. The fabricated BTO nanowires have a length of ~5 μm, which are capable to realize the pyro-driven ROS generation for degradation of organic dyes.

Total organic carbon (TOC) measurement was performed to monitor the total amount of organic carbon in the Indigo Carmine degradation experiment. As the reduction of TOC reflects the extent of degradation or mineralization of an organic species, the TOC value in the pyro-catalysis experiment was studied as a function of thermal cycles (Supplementary Fig. 9). The initial TOC value of the Indigo Carmine is 39.85 mg L$^{-1}$. After 9 thermal cycles with a temperature fluctuation of +5 °C, the TOC value decreased to 8.36 mg L$^{-1}$ (i.e., ~20% of the initial TOC value). The reduction of TOC confirms that color change was due to degradation of Indigo Carmine macromolecules (Eq. 5), rather than decolorization or decomposition.

Electron paramagnetic resonance (EPR) of the BTO nanowires pyro-catalyst using 5,5-dimethyl-1-pyrroline N-oxide (DMPO) as a spin trapper was used to detect •OH and •$O_2^-$ radicals that play a major role in the degradation of Indigo Carmine. After three thermal cycles at $\Delta T = 5\,°C$ and $\Delta T = 10\,°C$, the signals of both DMPO- •OH (Fig. 3h) and DMPO- •$O_2^-$ (Fig. 3i) were observed. The additional peaks in Fig. 3i are from the intermediate product DMSO- •$CH_3$ formed by the reaction of •OH and DMSO[47,48]. The signal of radicals was enhanced with the increase of temperature change, which implies an elevated number of radicals. Furthermore, level of radicals also increased with increasing

number of thermal cycles (Supplementary Fig. 10). The radicals created by PMN-PT via pyro-catalysis were also tested by the same process, the signals of EPR for both •OH and •$O_2^-$ demonstrated a strong dependence on the pyroelectric effect (Supplementary Fig. 11). The correlation of the radical signal with the temperature fluctuation and the number of thermal cycles confirms that the radical was derived from the pyroelectric effect of BTO nanowires.

Due to the complexity of the human oral environment resulting in saliva with many other metal ions as well as enzymes, artificial saliva was employed as a solvent for pyro-catalytic indigo carmine degradation in order to exclude the effect of saliva environmental complexity. The BTO nanowires were subjected to three cycles at a temperature fluctuation of +5 °C. It can be seen that the BTO nanowires exhibit excellent cycling stability in the artificial saliva environment (Supplementary Fig. 12), and their pyro-catalytic performance in this artificial saliva environment is almost the same as that in deionized water (Fig. 3c). In addition, the phase structure and morphology of the BTO nanowires themselves remains stable (Supplementary Fig. 13). This result provides strong support for the application of pyro-catalysis for tooth whitening.

## Tooth whitening demonstration based on pyro-catalysis

After a successful demonstration of pyro-catalytic generation of reactive species and the subsequent degradation of Indigo Carmine solution by these radicals, tooth whitening experiments were performed using BTO nanowires. Human teeth were soaked in a mixture of black tea, red wine, and blueberry juice for one week to simulate the staining of teeth caused by habitual food intake. The stained teeth were then placed in a BTO nanowire suspension at an initial temperature of 36 °C (to simulate the oral temperature), and pyro-catalysis experiments were performed at different temperature fluctuations. The photographs of the teeth treated under various thermal cycles at $\Delta T = -10, +10, +25\,°C$ were obtained using the same tooth and a standard greyscale card was used as a reference when the photographs were taken (Fig. 4a–c). It is obvious that the tooth enamel is significantly whitened after 2000 thermal cycles of pyro-catalysis (compare left-most and right-most photo in each sequence), which is consistent with the previous experimental results that the higher the temperature and the greater the temperature fluctuation the more effective the teeth whitening effect is. It should be noted that at $\Delta T = +25\,°C$, even the roots, which are the most deeply stained part of the teeth and the hardest part to deal with, were completely whitened (Fig. 4c). In contrast, at $\Delta T = +25\,°C$, the tooth whitening effect of the control (aqueous solution without BTO nanowires) was almost negligible (Fig. 4d).

Typically, sugars and amino acids in food will remain on the surface of tooth and interact with the components in saliva to form colored chromogens through complex chemical changes, so the process of tooth whitening is to degrade the chromogen into colorless small molecules. Under pyro-catalysis, continuous temperature cycles excite the pyroelectric property of BTO nanowires, continuously releasing screening charges to form active radicals. These radical species attack the stains on the tooth surface by oxidizing the multiple conjugated double bonds of the large organic molecules that generated the stains[49]. The Commission International De L'Eclairage (CIE-Lab) system was also used to quantitatively characterize the degree of tooth whitening. The CIELab system characterizes the change in tooth shade by three elements: $L$ (bright-dark), $a$ (red-green), and $b$ (blue-yellow)[50]. The comparison of tooth chromaticity before and after pyro-catalysis was measured for different treatment environments, and the values of $L$, $a$, $b$ are given in Fig. 4e–g, respectively. When the pyro-catalyst was present, the brightness $L$ of the teeth was significantly enhanced, while the value of $a$ decreased significantly, and the value of $b$ changed slightly. Conversely, changes in the values of $L$, $a$, and $b$ for the teeth without BTO nanowires as the whitening agent are negligible.

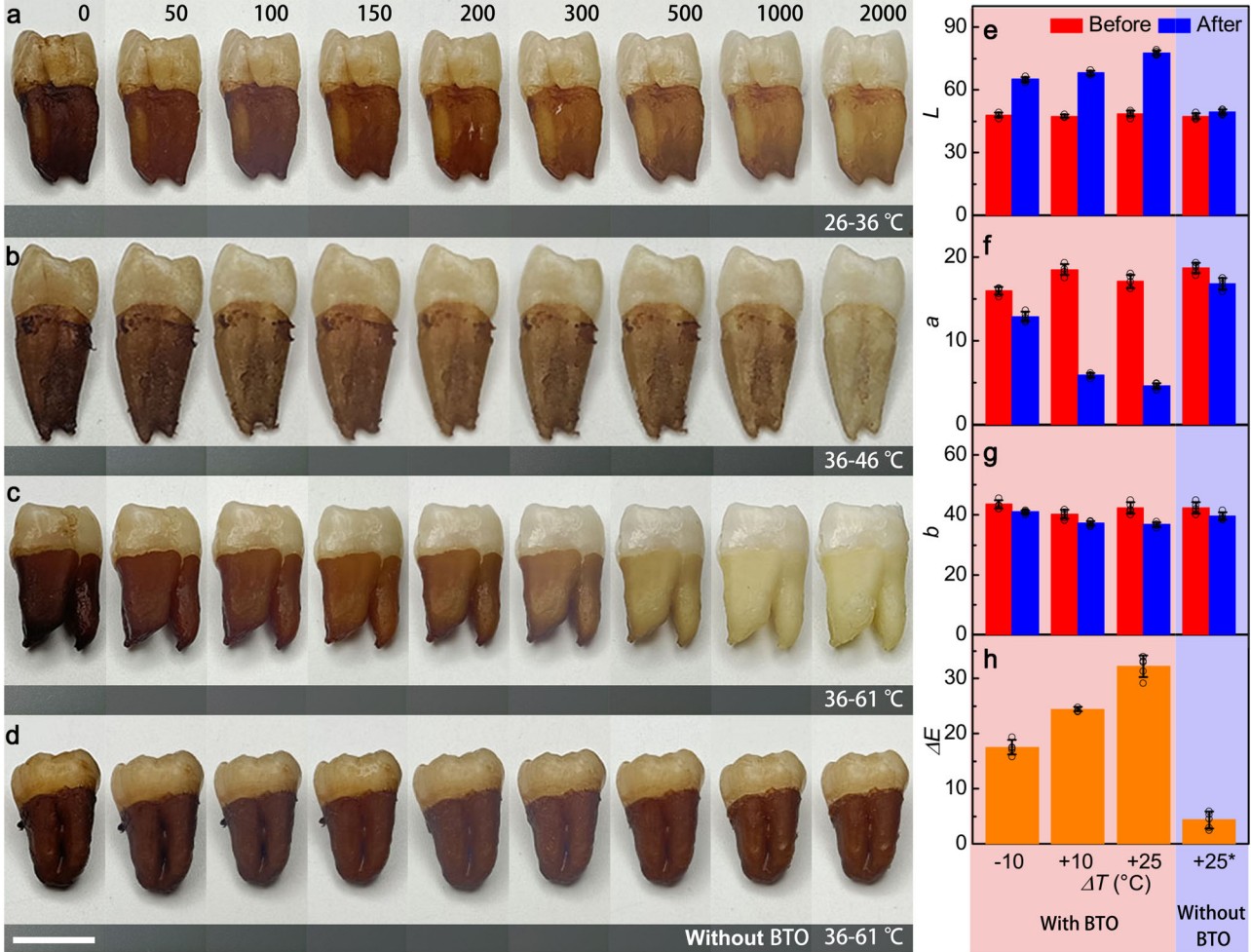

**Fig. 4 | Demonstration of tooth whitening based on pyro-catalysis effect.**
Photographs of teeth under treatment in turbid liquid of BTO nanowires with different temperature fluctuations **a-c** $\Delta T = -10, +10, +25 \,°C$, respectively.
**d** Photographs of teeth under treatment in pure water with a temperature fluctuation of $+25 \,°C$. These photographs are successive images of the same tooth.

Comparison of different temperature fluctuation on the tooth whitening levels demonstrated by CIELab results **e** luminance $L$, **f** color value of red–green axis $a$, **g** color value of blue-yellow axis $b$ and **h** color difference $\Delta E$ (+25* means without BTO nanowires). Scale bar is 1 cm. Data are presented as mean values ± SD (n = 5). Source data are provided as a Source Data file.

The chromaticity changes for the pyro-catalysis effect are more obvious with increased absolute temperature and increased temperature change. The $\Delta E$ calculated from Eq. (7) was used to further characterize the effect of whitening teeth[51].

$$\triangle E = \sqrt{\triangle L^2 + \triangle a^2 + \triangle b^2} \qquad (7)$$

Figure 4h shows the calculated $\Delta E$ for teeth treated in different environments. The $\Delta E$ for teeth treated in the environment containing the pyro-catalyst was about four times higher than for deionized water when $\Delta T = 25 \,°C$. Moreover, similar to the experimental results for the degradation of organic dyes there is an obvious correlation between the value of $\Delta E$ and both system temperature and the magnitude of temperature fluctuation. The tooth whitening procedure was also verified using PMN-PT single crystal powder as catalyst, with the PMN-PT samples showing a better tooth whitening effect after 2000 thermal than BTO nanowires (Supplementary Fig. 14).

**Pyro-catalysis performance of BTO-Gel**
After successful demonstration of tooth whitening by pyro-catalysis, we developed a practical delivery method. A photo-cured hydrogel was used as a carrier for the pyroelectric powders in preparation of composite gels to serve in the fabrication of medical braces. To verify

the retention of pyro-catalytic performance of the BTO nanowires in the composite gel form, degradation experiments of Indigo Carmine solution were repeated for the composite. As shown in the UV-VIS absorption curves of Fig. 5a, the gel with the composite pyroelectric material exhibited pyroelectric catalytic performance, which was comparable to that of the pyroelectric nanowires alone, confirming that the presence of the medical gel did not interfere with the pyroelectric catalytic performance. The pyro-catalytic performance stability of the composite gel was then tested by using the same gel for five sequential degradation experiments with Indigo Carmine solution. The composite gel showed no changes between successive cycles (Fig. 5b).

The ability of the pyroelectric composite gel to generate radicals was also characterized. The composite gel was placed in water with DMPO as the trapping agent and subjected to three thermal cycles. The DMPO- •OH signal detected at $\Delta T = 10 \,°C$ was approximately twice as high as that at $\Delta T = 5 \,°C$ (Fig. 5c). This result was consistent with the EPR test results for BTO nanowires (Fig. 3h–i). Furthermore, the signal of DMPO- •OH was detected at $\Delta T = 5 \,°C$ for multiple, successive thermal cycles (Fig. 5d), and the intensity of the DMPO- •OH signal increased with the increase of the number of thermal cycles, which proves that the radicals are continuously generated as the thermal cycles proceed. The detection of the DMPO- •O$_2^-$ signal performed in DMSO solution exhibited a similar trend (Fig. 5f–g), which further

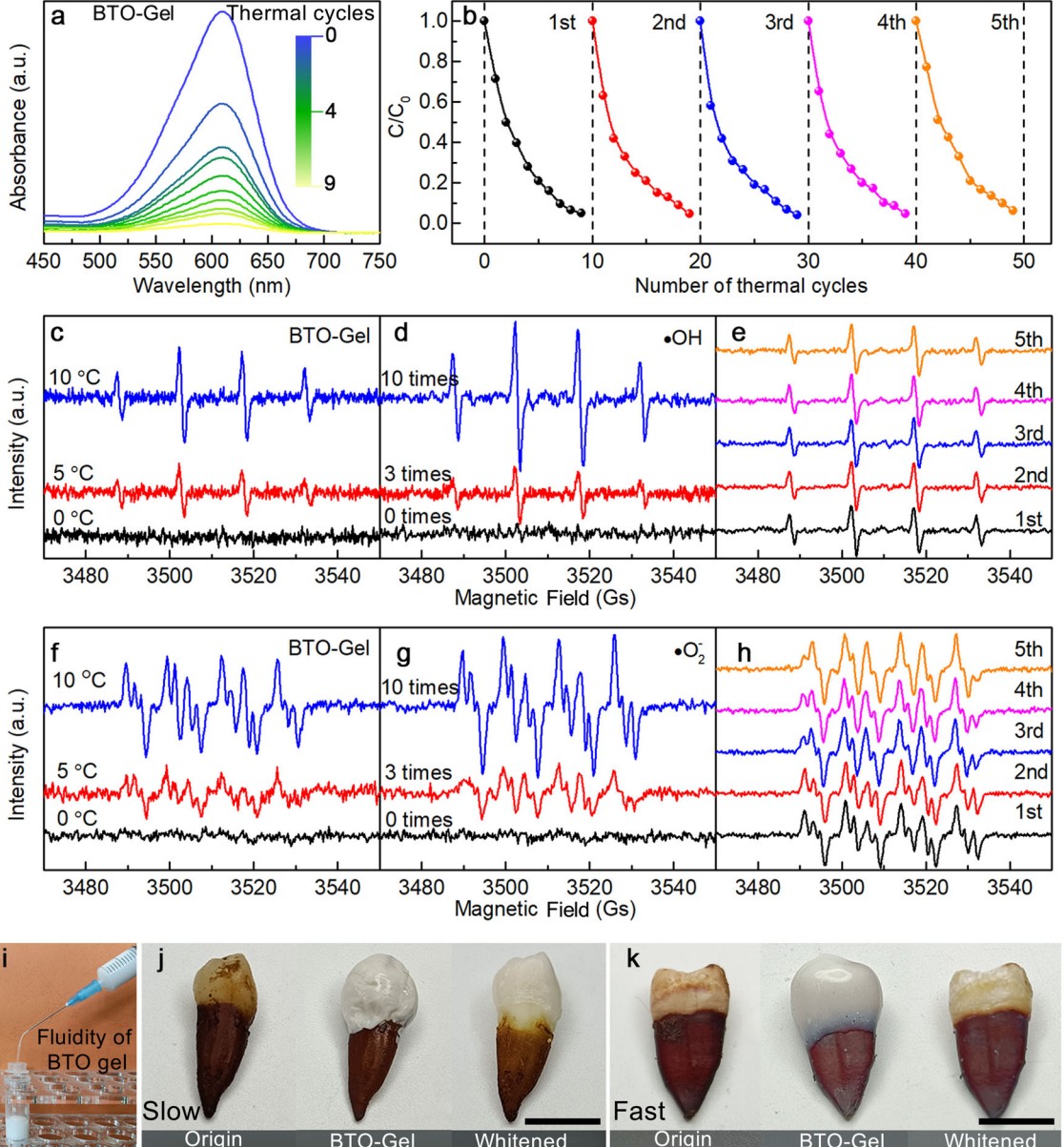

**Fig. 5 | Degradation properties of BTO-Gel. a** UV-Vis absorption spectra of Indigo Carmine solutions using BTO-Gel with a temperature fluctuation of +5 °C. **b** Cyclic stability of BTO-Gel degraded indigo solution. **c**–**h** Electron paramagnetic resonance spectra (EPR) of radical created by pyro-catalysis over different temperature range, different cycling times and the stability of radical creation. **i** The hydrogel shows excellent fluidity before curing even can be ejected from the syringe. Photographs of the stained tooth at original state, coated by BTO-gel and whited by BTO-gel at **j** slow and **k** fast heating rate. Slow rate designates heating-cooling time is 5 min, and fast designates heating-cooling time is 5 s. Scale bars are 1 cm. Source data are provided as a Source Data file.

confirms that the signals of the radicals generated in the composite gel come from the pyroelectric material contained therein.

To further characterize the stability of the pyro-catalysis of the composite gels, pyroelectric radical generation was tested after subsequent degradation cycles. Each test was performed at $\Delta T = 10$ °C for three thermal cycles, and the results (Fig. 5e, h) showed that after five dye degradation experiments, the composite gels still exhibit excellent radical production of both •OH and •O$_2^-$. The fluidity of the BTO gel is demonstrated in Fig. 5i. The excellent fluidity before curing allows the gel to be molded into any shape, making it possible to prepare intricate dental braces, but also allows for customization for a pinpoint treatment of a single tooth. A stained tooth was placed in a mold and BTO gel was injected into the mold to completely encapsulate the enamel of the tooth. The gel was then cured for 15 min in UV light. Afterwards, the tooth was placed in water and subjected to 2000 thermal cycles at a temperature fluctuation of 25 °C with different heating rate (Fig. 5j, k).

It was evident that the gel-coated parts of the teeth were significantly whitened compared to the initial state, while the non-gel-coated parts showed no significant color change. In order to simulate more realistic oral temperature environment, the stained tooth with BTO gel was immersed into hot and cold side each for 5 s (Supplementary Movie 3). The tooth whitening effect is also obviously occurred, while in contrast, there was no perceptible change in the color of the one with 5 min, which agrees well with the pyro-catalytic dye degradation (Supplementary Fig. 7).

## Tooth structure characterization

Dental enamel, as the hardest tissue in the body, acts as a protective covering of teeth and can withstand a wide range of functional and non-functional loads[52]. Classical tooth whitening methods using peroxide are effective, but can cause adverse such as increased surface roughness, cracking, and enamel changes[53]. To evaluate the safety of

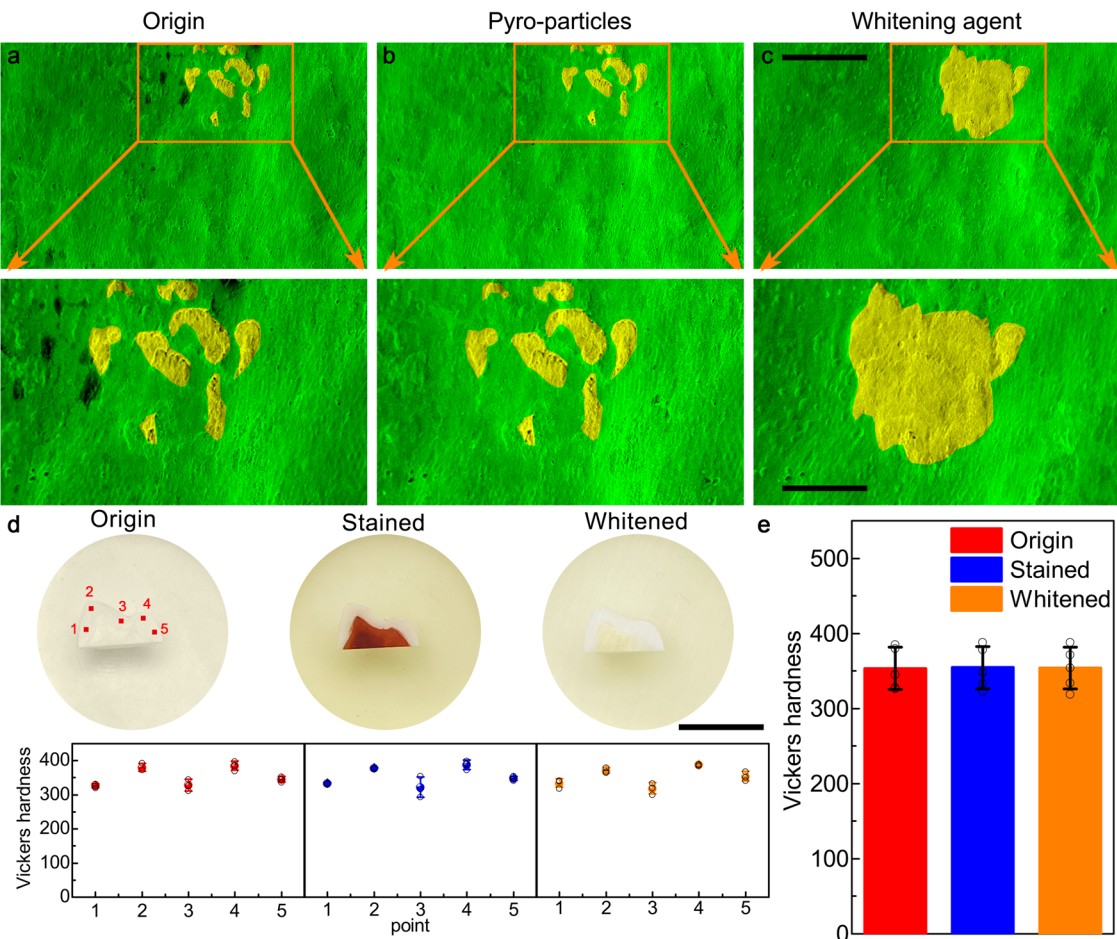

**Fig. 6 | Non-destructive characterization.** Scanning electron micrographs of the same area of tooth enamel **a** before whitening, **b** after whitening with BTO gel, and **c** after further whitening with commercial peroxide gel. **d** The Vickers microhardness of five points on the enamel of the same tooth in different states ($n = 3$) and **e** the comparison of the average microhardness of the enamel in different states ($n = 5$). Scale bar: **c** are 100 μm (top) and 50 μm (bottom), **d** is 1 cm. The experiments in **a**–**c** were repeated independently for three times with similar results. Data are presented as mean values ± SD. Source data are provided as a Source Data file.

the BTO composite gels, the microscopic morphology of tooth enamel before and after whitening with different whitening agents was examined. As shown in Fig. 6a, the same area of the same tooth was observed and recorded using scanning electron microscopy (SEM). The enamel surface of the tooth was rough and stained in the initial state, and after 2000 thermal cycles with BTO gel as the whitening agent, stains are removed from the tooth surface, and no damage was caused to the enamel owing to the gentle and continuous release of ROS (Fig. 6b). In contrast, enamel whitened with commercial tooth whitening gels showed significant and irreversible damage to the enamel due to the violent nature of the response caused by the dramatic release of ROS from high peroxide concentrations (Fig. 6c). To further verify that the pyroelectric-catalyzed tooth whitening technique does not cause damage to teeth microhardness of the tooth enamel was measured at five locations before and after whitening (Fig. 6d, e). In the as-received condition, the hardness at the five different positions is similar, with an average value of ~350 HV, indicative of a complete and healthy tooth. There is little to no change in the hardness across the tooth's surface between the as-received, the stained, and the pyro-catalytically whitened condition. This suggests that the pyroelectric gel does not cause mechanical damage to the enamel while whitening the tooth, much less cause any functional defects to the tooth.

To evaluate the biocompatibility of pyro-catalysis gel, fibroblast cells (L-929) were used to co-cultured with BTO nanowires and evaluated using the live/dead fluorescence staining and typical cell-counting kit 8 (CCK-8) assays. Fluorescence microscope images of

tissue cultures exposed to different BTO nanowires doses across a three-day period are presented in Fig. 7a-d. Since no significant cell viability decrease can be detected, even at the BTO nanowires dose of as high as $0.3\,g\,mL^{-1}$, the BTO nanowires can be confirmed as biocompatible. Figure 7e-g presents the results of the CCK-8 assay. The cell survival rate of all experimental groups was above 70%, showing that the BTO nanowires have no cytotoxicity to the fibroblast cells. As a further precaution, $Ba^{2+}$ leakage during the whitening process was also examined. The results showed no detectable $Ba^{2+}$ creation after 2000 thermal cycles (Supplementary Fig. 15).

## Discussion

In conclusion, we have demonstrated a non-destructive, biocompatible and time-efficient tooth whitening strategy based on a BTO nanowires pyro-catalyst composite hydrogel. Ferroelectric tetragonal BTO nanowires with a length of ~5 μm were synthesized as a catalyzer by a hydrothermal method with self-polarization capability (Supplementary Fig. 16). Degradation of Indigo Carmine was examined by exposure to BTO nanowires turbid liquid under different temperature fluctuations. The degradation rate of organic dyes using these catalyzers verified the catalysis effect was a result of the pyroelectricity since the effect was noted in both pyroelectric BTO nanowires and PMN-PT. Furthermore, teeth stained with different agents were completely whitened with BTO nanowires turbid liquid after 2000 thermal cycles with a temperature fluctuation of 25 °C, as determined by CIELab characterization. Additionally, a prototype tooth whitening dental

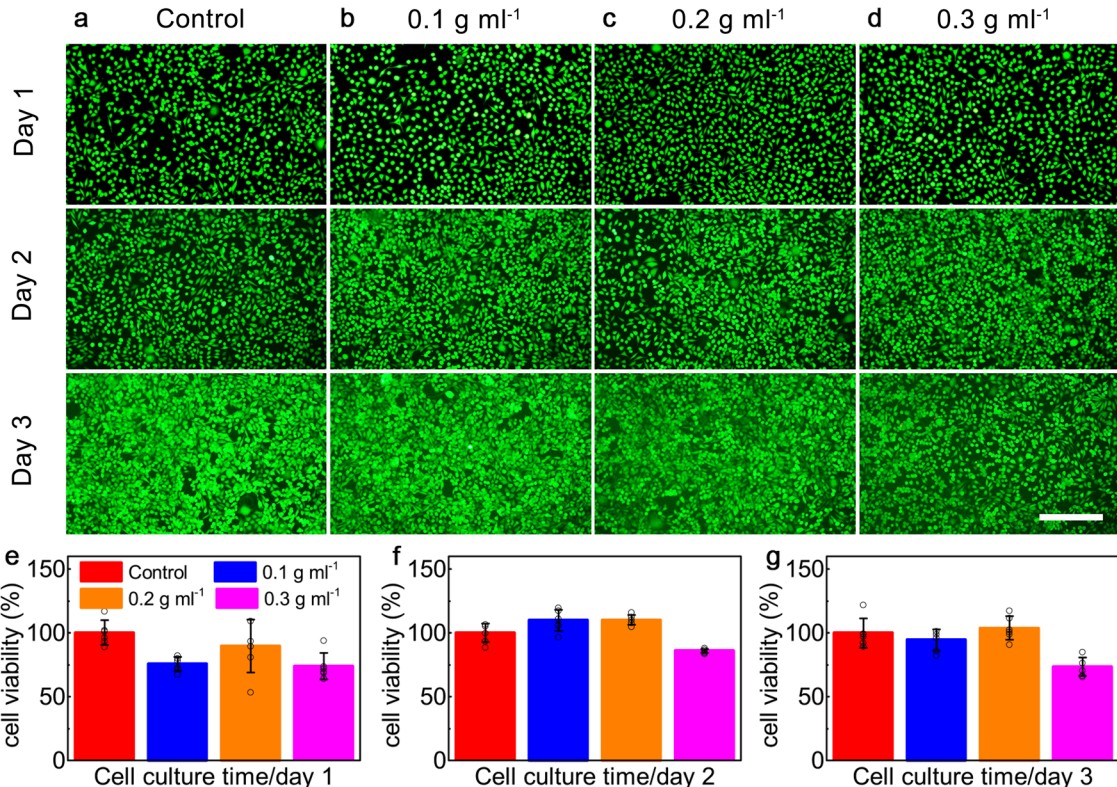

**Fig. 7 | Cytotoxicity characterization. a–d** The fluorescence microscope images of L-929 cells exposed to BTO nanowires with different concentrations for 1,2 and 3 days respectively. Dead cells appear red, while living cells appear green. **e-g** The viability of L-929 cells exposed to BTO nanowires with different concentrations for 1, 2 and 3 days measured by CCK-8 assay. Scale bar is 150 µm. The experiments in **a-d** were repeated independently for three times with similar results. Data are presented as mean values ± SD (**n** = 6). Source data are provided as a Source Data file.

brace using BTO nanowires combined with light-cured hydrogel shows excellent chemical and structural stability. The pyro-catalysis-based tooth whitening braces were also demonstrated using a stained tooth coated by BTO gels. Our results indicate that the pyro-catalysis-based tooth whitening via BTO nanowires is non-destructive to the tooth enamel and biocompatible without cytotoxicity. The concepts and results strongly highlight the bright prospects of pyroelectric materials used for tooth whitening, or even oral sterilization (Supplementary Fig. 17). This strategy can be conveniently implemented during our daily oral activities (e.g., drinking, breathing, talking, exercising, etc.) without extra time-consuming and additional equipment.

## Methods
### Materials preparation
$BaTiO_3$ nanowires were synthesized by a two-step hydrothermal method using $H_2Ti_3O_7$ nanowires as templates. First, 1.5 g of P25 particles were added into 60 mL of NaOH (10 M) aqueous solution and stirred for 2 h. This dispersion was poured in a Teflon autoclave (100 mL) and heated at 180 °C for 12 h to form $Na_2Ti_3O_7$ nanowires. The products were collected by centrifugation and washed four times with deionized water and ethanol before drying at 80 °C for 12 h. The precipitates were soaked in 0.2 mol $L^{-1}$ HCl solution with slow stirring for 4 h to form $H_2Ti_3O_7$ nanowires.

In the second step, mixtures of $H_2Ti_3O_7$ and $Ba(OH)_2 \cdot 8H_2O$ were hydrothermally reacted to form $BaTiO_3$ nanowires. In detail, 0.2575 g $H_2Ti_3O_7$ was added into 60 ml $Ba(OH)_2 \cdot 8H_2O$ aqueous solution (0.05 mol $L^{-1}$) and ultrasonically treated for 30 min, after which the mixture was vigorous stirred for 1 h. Finally, the mixture was transferred to a Teflon autoclave reactor (100 mL) and heated at 210 °C for 180 min. The final products were washed with 0.2 mol $L^{-1}$ HCl solution,

deionized water, and alcohol before drying at 80 °C for 12 h. The chemicals used are analytically pure

### Structural characterization
The phase structure of $BaTiO_3$ nanowires were determined by X-ray diffractometry (XRD, Bruker D8) with Cu Kα radiation (λ = 1.5406 Å, 2θ = 20°–60°). Microstructures were characterized by a field-emission scanning electron microscopy (SEM, ZEISS Merlin), a transmission electron microscopy (TEM) and a field-emission high-resolution transmission electron microscopy (HRTEM, FEI Tecnai G20). The sample was dispersed in ethanol, and a drop of solution was deposited onto a Si (100) substrate to allow the ethanol solvent to evaporate. A piezoresponse force microscopy (PFM, MFP-3D) was used to characterize the piezoelectric performance of the materials. The optical absorption spectra of Indigo Carmine molecules in the centrifugation was measured to determine the concentration of dye solutions by a Shimadzu UV-3600 UV−VIS−NIR spectrophotometer.

### Pyro-catalytic effect test
The pyro-catalytic performance of the present $BaTiO_3$ nanowires was examined by the degradation of organic dyes in aqueous solutions with repetitive cooling-heating cycles between different temperature fluctuations. The temperature fluctuations were controlled by an automatic heating-cooling circulation machine via, and there is no interval between each heating or cooling cycles. The heating-cooling rate was determined by stirring hotplate with infrared temperature recording system. Indigo Carmine was selected as the target pollutant, and 50 mg BTO nanowires were dispersed in 50 mL of 10 mg $L^{-1}$ Indigo Carmine aqueous solution and 1%vol. $H_2O_2$ with a concentration of 30% was added as co-catalyst. The obtained solutions were stirred for

30 min to reach an adsorption-desorption equilibrium between BTO nanowires and Indigo Carmine molecules. 3 mL samples of the suspension were periodically collected and centrifuged to remove the catalysts. Finally, the concentration of Indigo Carmine remained in clean solution was determined by a UV-Vis spectrophotometer.

## Detection of reactive species

The reactive species created by the pyro-catalyst were detected by the electron paramagnetic resonance (EPR) technique with a Bruker A200 spectrometer. First, 10 mg pyro-catalyst were dissolved in 10 mL of deionized water or 10 mL of dimethyl sulfoxide for •OH and •$O_2^-$ detection, respectively. 200 μL solution was taken out and 20 μL of 5,5-dimethyl-1-pyrroline N-oxide (DMPO) was added into the solution. The reactive species were detected immediately after cooling–heating cycles for 0, 3, and 10 times with a temperature fluctuation of 5 °C, or after cooling-heating cycles of 3 times at a temperature fluctuation of 0, 5, 10 °C.

## Tooth whitening experiment

The extracted teeth (with informed consent by the donors) we choose were healthy and free of caries. Each tooth was washed with pure water immediately after removal, and soft tissues were scraped away. The teeth were soaked in 0.5% Chloramine-T solution (9.0 g sodium chloride and 5.0 g chloramine–trihydrate dissolved in 1000 ml distilled water). Before using, teeth were stained by mixture of black tea, blueberry juice, and wine for 1 week. After that, the teeth were washed using deionized water until the rinse water was clear. Finally, the teeth were dried using bibulous paper. The stained teeth were placed in beakers with 50 mL deionized water, 50 mL ubiquitous suspension (1 mg mL$^{-1}$). After stirring for 30 min, the beakers were subjected to 2000 cooling–heating cycles at 25 °C temperature fluctuations via immerseing into hot and cold side each for 5 s and 5 min, respectively. The chroma of the tooth enamel was periodically sampled by computer-aided shade matching (VITA Easyshade). In order to observe the influence on enamel after whitening, we observed an area on surface of a colored tooth by SEM before whitening. The tooth was treated by BTO-Gel and commercial tooth whitening gel successively. The same area on surface was observed by SEM after each whitening experiment. The experiments were approved by Institutional Review Board of Peking University School and Hospital of Stomatology.

## Cytotoxicity testing

Different concentrations of BTO nanowires were soaked in the basic Minimum Essential Medium (MEM) medium for 24 h at 37 °C in an incubator under an atmosphere containing 5% $CO_2$. The supernatant extracted by centrifuging at 1000 rpm for 5 min, was then mixed with 10% fetal bovine serum and 1% penicillin-streptomycin solution to coculture with cells. Fibroblast cells (L-929, provided by Shanghai Institute of Cell Biology, Chinese Academy of Sciences) were cultured to logarithmic phase and seeded in 48-well plates ($1.2 \times 10^4$ cells per well) and soaked in unprocessed complete medium (MEM, Procell, China) supplemented with 10% fetal bovine serum and 1% penicillin-streptomycin solution at 37 °C in an incubator under an atmosphere containing 5% $CO_2$. After a 24-hour incubation, each groups' medium was replaced by the extracted one. In addition, a control group was set. For each day, the samples were stained with Calcein/PI Cell Viability/ Cytotoxicity Assay Kit (5 μg mL$^{-1}$ each in PBS). After 15 min of staining, the samples were rinsed with PBS and imaged at the WU module using a fluorescence microscope (Olympus IX53, Japan). We used the Cell Counting Kit-8 (SpectraMax 190, Molecular Devices Corporation, USA) assay to evaluate the cell viability of each group after being statically cultured for 1, 2, and 3 days. Briefly, for each 100 μl of medium solution, 10 μl of CCK-8 solution (Meilunbio, China) was added into each well of the plate and incubated for 1.5 h. Finally, the cell viability was calculated according to the absorbance measured at 450 nm with a microplate reader.

## Data availability

The data that support the findings of this study are available from the corresponding author on request. Original data of all figures presented are provided in the source data file. Source data are provided with this paper.

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

## Acknowledgements

Y.J.W. acknowledges the National Natural Science Foundation of China (51790492 and 52072178) and the Fundamental Research Funds for the Central Universities (30918012201), X.L.D. acknowledges the National Natural Science Foundation of China (81991505), X.H.Z. acknowledges the National Natural Science Foundation of China (82022016) and the National Key Research and Development Program of China (2021YFB3800800), X.G.L. acknowledges the National Natural Science Foundation of China (U21A2066), Y.W. acknowledges Postgraduate Research & Practice Innovation Program of Jiangsu Province (KYCX20_0272).

## Author contributions

Y.J.W., X.H.Z. and X.L.D. conceived this work and designed the experiments; Y.J.W., Y.W., S.H.W., Y.Z.M., D.J.L., Y.Y.B. and G.L.Y. performed the experiments; Z.L. performed the simulation; the data analysis was performed by Y.J.W., Y.W., X.H.Z., X.G.L. and X.L.D.; Y.W. and Y.J.W. wrote the manuscript. All authors reviewed and commented on the manuscript.

## Competing interests

The authors declare no competing interests.
