## [Peer Review File · Nature Communications]

REVIEWER COMMENTS

Reviewer #1 (Remarks to the Author):

In this manuscript (NCOMMS-21-29906A-Z), titled by "Pyro-catalysis for tooth whitening via oral temperature fluctuation" by Prof. Wang and etc., pyroelectricity was used for pyrocatalytic tooth whitening. Obviously, it is an interesting and novel work. However, there are lots of issues needed to be claimed before accepted by Nature Communications, which were listed as follows:

1. What's the mechanism of action of H₂O₂ in the Supplementary Fig. 2? The pyrocatalysis becomes obvious with the addition of H₂O₂, is it due to the synergistic effect of pyrocatalysis and Fenton-like catalytic oxidation reaction?

2. It seems that the similar pyrocatalytic degradation of PMN-PT in Supplementary Fig. 4, which was previously mentioned in the supplementary materials of Nat. Commun. 9, 1-8 (2018). Therefore, it had better delete Supplementary Fig. 4.

3. The two videos are the cold-hot setup and the infra-imaging of drinking cold and hot water, respectively. Please provide the videos of obvious pyrocatalysis experiment displayed the results in the manuscript?

4. In the Fig. 3 and Supplementary Fig. 6 and 7, the unit of magnetic field should be "Gs", not "G". The different cycling times exhibiting different curves should be labeled in the Supplementary Fig. 6.

5. The pyroelectric performance of poled and unpoled BaTiO₃ catalyst should be provided in this work.

6. In the equation 1, the multiplication "." should be added between these physical quantities.

7. What's the final product of the organic matter on the tooth after pyrocatalytic whitening? Decolorization, decomposition or degradation? It's better to do the TOC measurement.

8. Please provided the scale line mark of the morphology characterization in Fig. 2.

9. Is the nanowire BTO broken in the pyrocatalytic experiments? Please provide the comparison before and after pyrocatalysis with the different microstructure and morphology of BTO catalyst.

10. How about the effect of the size and shape of BTO catalyst on the pyrocatalysis?

Reviewer #2 (Remarks to the Author):

This manuscript aimed at to study a novel tooth-whitening strategy based on pyroelectric material (BaTiO₃ hydrogel)-decorated braces as well as the impact of oral temperature fluctuations. And the resultant BaTiO₃ nanowires exhibited satisfactory pyro-catalytic activities for the degradation of Indigo Carmine pollutants at different temperature ranges. Interestingly, a self-made dental brace with BaTiO₃ hydrogel could significantly whiten teeth while exhibiting remarkably therapeutic biosafety and sustainability. This work revealed the practical application potential of pyroelectric materials in the field of tooth whitening. It's reasonable from the view of technique. Following issues can be considered to make this work better:

Specific comments are presented as follows:

1. In Introduction, more research developments on the application of the host material BaTiO₃ in pyroelectric-catalysis should be added.

2. In section "Indigo Carmine degradation based on pyro-catalysis", It says that "17 thermal cycles were required for a 95% degradation of Indigo Carmine at 26-36 °C ($\Delta T = -10$ °C), while only 6 thermal cycles were required for the same degradation of Indigo Carmine at 36-46 °C ($\Delta T = 10$ °C)." This result is not reflected in Supplementary Fig. 4.

3. P11, the fifth line in the third paragraph, "The total charges induced by the pyroelectricity at a rapid heating rate.", thereinto, how to reflect the charge recombination and release efficiency caused by pyroelectricity? It can be further discussed.

4. In this work, it was relatively scarce in the characterization and analysis of materials. As we know, the human oral environment is complex, in addition to exploring the recyclable activity of the BaTiO₃ nanowires, it is also necessary to explore the stability of the resultant material in a variety of simulated oral environments. At the same time, it is necessary to characterize and analyze the structure of the material before and after the reaction.
5. The composition of oral saliva is complex, whether there are some inorganic ions (e.g. K⁺, Na⁺, Ca⁺) and active enzyme species that compete for the charge generated by the pyroelectricity, disturb the trade-off between charge and ROS generation; secondly, the oral saliva contains lysozyme and thiocyanate ion that play a role in oral sterilization, whether cyanide ion can achieve the synergistic bactericidal effect with ROS in this simulated environment, it needs to be further explored.
6. What is the specific role of H₂O₂ in the pro-catalytic effect test? No specific explanation was given in the manuscript. In addition, we know that •O₂⁻ will be further reduced to generate H₂O₂. Whether H₂O₂ will be generated during the pyroelectric-catalysis process can be further discussed.
7. In the Fig. 5g-h, it is obvious that in addition to the EPR peak of •O₂⁻, there are other split peaks generated, indicating the generation of other derived radical intermediates (e.g. •CH₃, •CHO), which can be further explored. And it is best to supplement the corresponding mechanism study.

Dear Dr. Robert Guillatt and referees,

We are truly grateful to your critical comments and thoughtful suggestions. Based on these comments and suggestions, we have made careful modifications on the original manuscript. All changes made to the text are in red color. We hope the new manuscript will meet your journal's standard. Below you will find our point-by-point responses to the reviewers' comments/questions below:

Reviewer #1

In this manuscript (NCOMMS-21-29906A-Z), titled by "Pyro-catalysis for tooth whitening via oral temperature fluctuation" by Prof. Wang and etc., pyroelectricity was used for pyrocatalytic tooth whitening. Obviously, it is an interesting and novel work. However, there are lots of issues needed to be claimed before accepted by Nature Communications, which were listed as follows:

1. What's the mechanism of action of H₂O₂ in the Supplementary Fig. 2? The pyrocatalysis becomes obvious with the addition of H₂O₂, is it due to the synergistic effect of pyrocatalysis and Fenton-like catalytic oxidation reaction?
2. It seems that the similar pyrocatalytic degradation of PMN-PT in Supplementary Fig. 4, which was previously mentioned in the supplementary materials of Nat. Commun. 9, 1-8 (2018). Therefore, it had better delete Supplementary Fig. 4.
3. The two videos are the cold-hot setup and the infra-imaging of drinking cold and hot water, respectively. Please provide the videos of obvious pyrocatalysis experiment displayed the results in the manuscript?
4. In the Fig. 3 and Supplementary Fig. 6 and 7, the unit of magnetic field should be "Gs", not "G". The different cycling times exhibiting different curves should be labeled in the Supplementary Fig. 6.
5. The pyroelectric performance of poled and unpoled BaTiO₃ catalyst should be provided in this work.
6. In the equation 1, the multiplication "·" should be added between these physical quantities.

7. What's the final product of the organic matter on the tooth after photocatalytic whitening? Decolorization, decomposition or degradation? It's better to do the TOC measurement.

8. Please provide the scale line mark of the morphology characterization in Fig. 2.

9. Is the nanowire BTO broken in the photocatalytic experiments? Please provide the comparison before and after photocatalysis with the different microstructure and morphology of BTO catalyst.

10. How about the effect of the size and shape of BTO catalyst on the photocatalysis?

Comment 1: What's the mechanism of action of H₂O₂ in the Supplementary Fig. 2?

The photocatalysis becomes obvious with the addition of H₂O₂, is it due to the synergistic effect of photocatalysis and Fenton-like catalytic oxidation reaction?

Response: Thank you for this comment. It is clear that the addition of H₂O₂ significantly accelerated the degradation of indigo carmine based on the experimental results. It has been shown that hydrogen peroxide is involved in the catalytic process as an intermediate product in photocatalysis (see Fig. 1 in response letter), and the addition of hydrogen peroxide to the reaction system can induce a Fenton-like effect to enhance the catalytic rate for photocatalysis.

Fig.1 Photocatalytic reaction mechanisms (*Applied Catalysis B: Environmental*, 2020, 272: 118970.)

Analogous to the photocatalysis, we designed experiment to clarify the mechanism of action of H₂O₂ (Supplementary Fig. 3). We monitored the concentration of H₂O₂ in the solution using liquid chromatography during the pyro-catalysis process. The results show that the concentration of H₂O₂ did not change throughout the pyro-catalysis

process (Supplementary Fig. 3), so we concluded that there was both consumption and generation of H_2O_2 during the pyro-catalysis process. The added small amount of H_2O_2 combine with the charge released from the pyroelectric material, which then release free radicals in a Fenton-like reaction. Hence the pyro-catalysis reaction was accompanied by the generation of H_2O_2 , thus achieving a balance between generation and consumption of H_2O_2 .

Additionally, DPD-POD method was further employed to clarify the mechanism of action of H_2O_2 . The related results were shown in Supplementary Fig. 4 (Page 5).

Supplementary Fig. 4 **a** Typical absorption spectra obtained by testing different concentrations of H_2O_2 using the DPD-POD method. **b** The linear relationship between the H_2O_2 concentration and the peak of the absorption spectrum at 551 nm can be used to calculate the H_2O_2 concentration in the unknown solution. **c** the absorption spectra of H_2O_2 concentrations created by BTO during pyro-catalysis. **d** The concentration of H_2O_2 in the reaction system with and without BTO after a specific number of thermal cycles.

H_2O_2 detection. The concentration of H_2O_2 created during pyro-catalysis process was measured by a DPD-POD method. Briefly, 50mg of BTO was dispersed in 50ml

of water, and after a certain number of thermal cycles, samples of the liquid were centrifuged and 1 mL of the supernatant was added to a mixture of 3 mL of phosphate buffer (0.5 M, pH = 6), 6 mL of water, 0.05 mL of N,N-diethyl-p-phenylenediamine sulfate (DPD, 10 mg mL⁻¹) and 0.05 mL of peroxidase (POD, 1 mg mL⁻¹). After 30 s, the H₂O₂ concentration was measured at 551 nm on a UV–vis spectrophotometer.

DPD-POD method was further employed to verify the role of small amount addition of H₂O₂ during pyro-catalysis process^{1,2}. First, the linear relationship between the peak of the absorption spectrum at 551 nm and concentrations of added H₂O₂ indicates the validity of the DPD-POD method, as shown in Supplementary Fig. 4 a and b. It can be seen that the generation of H₂O₂ increase with the number of thermal cycles and about 300 μM of H₂O₂ was produced in the reaction system after 30 hot and cold cycles at a temperature fluctuation of +20 °C, as shown in Supplementary Fig. 4 a and b.

The discussion can be found in main text at page 10, and we also added refs. 39 and 40.

“In contrast, the degradation of Indigo Carmine was minimal when BTO nanowires were used as catalyst or when BTO nanowires were not added (Supplementary Fig. 2). The concentration of H₂O₂ monitored by liquid chromatography shows no change during the pyro-catalysis process (Supplementary Fig. 3), while the DPD-POD experimental results indicate the generation of 300 μM of H₂O₂ in the reaction system after 30 hot and cold cycles at a temperature difference of +20 °C (Supplementary Figs. 4). Anomalous to photo-catalysis, it can be inferred that H₂O₂ is as an intermediate product and remains an equilibrium state between consumption and generation during the pyro-catalysis process. The charges released from the pyro-catalytic BTO nanowires was first combined with the addition of H₂O₂, and subsequently react further into reactive radicals in a Fenton-like reaction^{39,40}. The addition of small amount of H₂O₂ can accelerate the pyro-catalytic process and improve the efficiency. These comparative experiments unambiguously verify that the degradation of Indigo Carmine due to the catalysis effect is strongly associated with the pyroelectricity of the nanowires.”

Comment 2: It seems that the similar pyrocatalytic degradation of PMN-PT in Supplementary Fig. 4, which was previously mentioned in the supplementary materials of Nat. Commun. 9, 1-8 (2018). Therefore, it had better delete Supplementary Fig. 4.

Response: Thank you for your suggestion. It is true that, as you mentioned, PMN-PT has been used to demonstrate pyro-catalysis effect in previous publications, but in this work, the pyro-catalytic degradation of indigo carmine by PMN-PT can effectively illustrate the relationship between pyro-catalysis effect and pyroelectric coefficient, and moreover, the supplementary results are consistent with the data in Supplementary Fig. 10. The author chose to keep it as Supplementary information. Moreover, some descriptions of the figure and related reference (Nat. Commun. 9, 1-8, 2018) have been added in Supplementary page 6.

“PMN-PT single crystal powder with high pyroelectric properties was used for comparative pyro-catalytic degradation experiments. Single crystals were first crushed and then poled using corona poling method³. The same degradation of indigo carmine using PMN-PT powders were performed as BTO nanowires at different temperature fluctuation. The results show that PMN-PT single crystal powder required fewer thermal cycles and the organic dyes can be degraded more completely, which is consistent with previous reports⁴.”

Comment 3: The two videos are the cold-hot setup and the infra-imaging of drinking cold and hot water, respectively. Please provide the videos of obvious pyrocatalysis experiment displayed the results in the manuscript?

Response: Thank you for this suggestion. We have provided a typical video of the entire process of degrading indigo carmine with BTO nanowires in Supplementary Video 1, and the corresponding text is in page 10: “Impressively, at temperature fluctuation of $\Delta T= 20$ °C, only three thermal cycles are required to degrade the Indigo Carmine (Supplementary Video 1).”

Comment 4: In the Fig. 3 and Supplementary Fig. 6 and 7, the unit of magnetic field should be “Gs”, not “G”. The different cycling times exhibiting different curves should be labeled in the Supplementary Fig. 6.

Response: Thanks a lot for your careful review, the units in the Fig. 3, Fig. 5, Supplementary Fig. 9 and Supplementary Fig. 10 has been carefully checked and modified, and a specific number of cycles was added to Supplementary Fig. 9 and Supplementary Fig. 10 to obtain a more intuitive result.

In the main text on page 9

Fig. 3 Degradation properties of pyro-catalysis. UV-Vis absorption spectra of Indigo Carmine solutions with respect to temperature fluctuations **a-f** $\Delta T = -10, -5, +5, +10, +15, +20$ °C. **g** the pseudo-first-order reaction kinetics of different temperature fluctuations. Electron paramagnetic resonance spectra (EPR) of radical **h** $\bullet\text{OH}$ and **i** $\bullet\text{O}_2^-$ created by pyro-catalysis over different temperature range.

In the main text on page 16

Fig. 5 Degradation properties of BTO-Gel. a UV-Vis absorption spectra of Indigo Carmine solutions using BTO-Gel with a temperature fluctuation of +5 °C. **b** Cyclic stability of BTO-Gel degraded indigo solution. **c-h** Electron paramagnetic resonance spectra (EPR) of radical created by pyro-catalysis over different temperature range, different cycling times and the stability of radical creation. **i** The hydrogel shows excellent fluidity before curing even can be ejected from the syringe. Photographs of the stained tooth at original state, coated by BTO-gel and whitened by BTO-gel at **j** slow and **k** fast heating rate. Slow rate designates heating-cooling time is 5 minutes, and fast designates heating-cooling time is 5 seconds. Scale bars are 1 cm.

In Supplementary information on page 10

Supplementary Fig. 9 Electron paramagnetic resonance spectra (EPR) of radical **a** $\bullet\text{OH}$ and **b** $\bullet\text{O}_2^-$ created by pyro-catalysis over different cycling times.

In Supplementary information on page 11

Supplementary Fig. 10 Electron paramagnetic resonance spectra (EPR) of radical $\bullet\text{OH}$

and $\bullet\text{O}_2^-$ created by PMN-PT over **a-b** different temperature range and **c-d** different cycling times.

Comment 5: The pyroelectric performance of poled and unpoled BaTiO₃ catalyst should be provided in this work.

Response: Thank you for your suggestion. Pyro-catalysis experiments were carried out using poled and unpoled BTO nanowires respectively. The results are shown in the Supplementary Fig. 15. It can be seen the pyro-catalytic performance of the poled BTO nanowires was decreased related to the unpoled BTO nanowires (or as-grown). It is highly possible that the as-grown BTO nanowires are mainly along the [001] direction, which is the highest pyroelectric coefficient direction for the tetragonal BTO. The grown nanowires have highly self-poled along the length direction. However, after being poled using an applied electric field, the polarization along the length direction was disrupted and rearranged leading to a decrease of pyroelectric properties. The same experiment was carried out using PMN-PT single crystal powder. In contrast to BTO nanowires, the catalytic performance of PMN-PT powder after poling was greatly improved for PMN-PT powder with randomly distributed polarization direction. These comparative experiments unambiguously verify that the pyro-catalysis performance is determined by the pyroelectricity (or degree of polarization), which is similar to the piezo-catalysis³.

Supplementary Fig. 15 UV-Vis absorption spectra of Indigo Carmine solutions using **a** poled, **b** unpoled BTO and **c** poled and **d** unpoled PMN-PT with a temperature fluctuation of +5 °C

Comment 6: In the equation 1, the multiplication “·” should be added between these physical quantities.

Response: Thank you for your careful examine. In the main text on page 11. The equation 1 has been changed into

$$\Delta Q = p \cdot A \cdot \Delta T \quad (1)$$

Comment 7: What’s the final product of the organic matter on the tooth after pyrocatalytic whitening? Decolorization, decomposition or degradation? It’s better to do the TOC measurement.

Response: Thank you for your critical suggestion. Since it is very difficult to test the change in the total amount of organic matter on the tooth surface during the catalytic

process, and the exogenous staining of teeth mainly comes from pigments in food, we aimed our tests at the degradation process of indigo carmine solution. TOC measurement was performed to determine the change of total organic matter in the indigo carmine solution during the catalytic process (Supplementary Fig. 8). The results show that the TOC value in the solution decreased throughout the catalytic process, indicating that the organic matter in the solution decreased and was converted into CO₂ and H₂O. Thus, it can be inferred that the organic matter on the tooth surface was degraded rather than discolored or decomposed during the tooth whitening process.

Supplementary Fig. 8 Total organic carbon (TOC) removal during Indigo Carmine pyro-catalysis.

The related discussion on the TOC experiment can be found in the main text on page 12.

“Total organic carbon (TOC) measurement was performed to monitor the total amount of organic carbon in the Indigo Carmine degradation experiment. As the reduction of TOC reflects the extent of degradation or mineralization of an organic species, the TOC value in the pyro-catalysis experiment was studied as a function of thermal cycles (Supplementary Fig. 8). The initial TOC value of the Indigo Carmine is 39.85 mg L⁻¹. After 9 thermal cycles with a temperature fluctuation of +5 °C, the TOC value decreased to 8.36 mg L⁻¹ (i.e., ~20% of the initial TOC value). The reduction of TOC confirms that color change was due to degradation of Indigo Carmine

macromolecules (Eq.5), rather than decolorization or decomposition.”

Comment 8: Please provided the scale line mark of the morphology characterization in Fig. 2.

Response: Thanks a lot for this careful review. We have provided a scale for each of the morphology characterization in Figure 2. In the main text on page 6.

Fig. 2 Microstructural and morphology characterization. a X-ray diffraction pattern of the BTO nanowires. **b** Room-temperature Raman spectra of the hydrothermal BTO nanowires. **c** Scanning electron microscope image of BTO nanowires, **d** Transmission electron microscope, **e** high-resolution transmission electron microscope images and **f**

selected area electron diffraction patterns of the BTO nanowires, and **g-i** corresponding EDX element mapping of Ba (red), Ti (blue), and O (yellow) in BTO nanowires. PFM results of BTO nanowires **k** topography image; **l** vertical amplitude image; **m** vertical phase image and **n** piezoelectric hysteresis loop. Scale bar: **c** is 10 μm , **d** is 1 μm , **e** is 5 nm, **f** is 5 1/nm, **g, h, i** and **j** are 1 μm , **k, l** and **m** are 500 nm.

Comment 9: Is the nanowire BTO broken in the pyrocatalytic experiments? Please provide the comparison before and after pyrocatalysis with the different microstructure and morphology of BTO catalyst.

Response: Thank you for your critical comment. The XRD and SEM tests were performed on the BTO nanowires before and after the catalytic experiments in order to verify that the microstructure and morphology of the BTO nanowires before and after the pyro-catalysis experiments. It can be seen that the BTO nanowires before and after the catalytic experiments match the standard PDF cards, and the morphology of BTO does not change, which corresponds to the outstanding cycling stability of BTO nanowires.

The comparison of BTO before and after pyro-catalysis has been added as Supplementary Fig. 12 and related discussion can be found in main text on page 13.

“...In addition, the phase structure and morphology of the BTO nanowires themselves remains stable (Supplementary Fig. 12)....”

Supplementary Fig. 12 a X-ray diffraction pattern of the BTO nanowires before and after pyro-catalysis, and scanning electron microscope image of BTO nanowires **b** before and **c** after pyro-catalysis.

Comment 10: How about the effect of the size and shape of BTO catalyst on the pyrocatalysis?

Response: Thanks a lot for your critical comment. The effect of the size and morphology of BTO on the pyro-catalytic performance has been previously investigated by both experiment and simulation.

Figure 3. STEM and element mapping images of BaTiO₃ samples: (a1–a4) BTO NWs, (b1–b4) BTO NPs-1, and (c1–c4) BTO NPs-2.

BTO catalyst with different size and shape. [ACS Appl Mater Interfaces 2018, 10, 37963.]

Experimentally, the catalytic performance of nanoscale materials is usually associated with the large specific surface area. In ref. 27, “...Since the BTO nanowires exhibit preferential grain growth along the polar axis, the area of reactive polar surface decreases as the length of nanowires increases, and a lower pyroelectric catalytic efficiency can be expected. In practice, however, the catalytic efficiency of nanowires is higher than that of nanoparticles. For these reasons **one may exclude the nanoscale size effect from the dominant effect on pyroelectric catalytic efficiency**. Instead, the morphology dependence of pyroelectric potential was considered as the main reason. [ACS Appl Mater Interfaces 2018, 10, 37963.]. This indicates that the effect of the size of the nanomaterials is not dominant in pyro-catalysis, but rather the pyroelectric

potential that can be generated by the material dominates the rate at which the catalytic reaction proceeds.

Pyro-catalysis performance of BTO with different size and shape. [*ACS Appl Mater Interfaces* **2018**, *10*, 37963.]

The simulation results also reveal a similar conclusion. Since the BTO nanowires grow preferentially along the polar axis, the nanowires have spontaneous polarization along the length direction. The results show that the BTO nanowires exhibit superior pyroelectric properties with higher pyroelectric potential relative to nanoparticles of smaller size and larger surface area.

Figure 8. (a) Selected temperature stage in the periodic thermal cycles, and the corresponding distribution of pyroelectric potential of BTO nanocrystals in the form of (b) particle in size of 100 nm × 100 nm × 100 nm, (c) particle in size of 200 nm × 200 nm × 200 nm, and (d) wire in size of 100 nm × 100 nm × 5000 nm.

Pyroelectric potential of BTO nanocrystals with different morphologies. [*ACS Appl Mater Interfaces* 2018, 10, 37963.]

Based on the reported result, we believe that the effect of size on the performance of pyro-catalysis is minimal, and the performance of pyro-catalysis primarily depends on how much pyroelectric potential can be generated. BTO nanowires have better pyro-catalysis performance due to their spontaneous polarization along the length direction, and can generate considerable pyroelectric potential, which is an advantage relative to other morphologies. This is the main reason why we chose BTO nanowires in our experiment.

Reviewer #2:

This manuscript aimed at to study a novel tooth-whitening strategy based on pyroelectric material (BaTiO₃ hydrogel)-decorated braces as well as the impact of oral temperature fluctuations. And the resultant BaTiO₃ nanowires exhibited satisfactory pyro-catalytic activities for the degradation of Indigo Carmine pollutants at different temperature ranges. Interestingly, a self-made dental brace with BaTiO₃ hydrogel could significantly whiten teeth while exhibiting remarkably therapeutic biosafety and sustainability. This work revealed the practical application potential of pyroelectric materials in the field of tooth whitening. It's reasonable from the view of technique. Following issues can be considered to make this work better:

Specific comments are presented as follows:

1. In Introduction, more research developments on the application of the host material BaTiO₃ in pyroelectric-catalysis should be added.
2. In section "Indigo Carmine degradation based on pyro-catalysis", It says that "17 thermal cycles were required for a 95% degradation of Indigo Carmine at 26-36 °C ($\Delta T = -10$ °C), while only 6 thermal cycles were required for the same degradation of Indigo Carmine at 36-46 °C ($\Delta T = 10$ °C)." This result is not reflected in Supplementary Fig. 4.
3. P11, the fifth line in the third paragraph, "The total charges induced by the pyroelectricity at a rapid heating rate.", thereinto, how to reflect the charge recombination and release efficiency caused by pyroelectricity? It can be further discussed.
4. In this work, it was relatively scarce in the characterization and analysis of materials. As we know, the human oral environment is complex, in addition to exploring the recyclable activity of the BaTiO₃ nanowires, it is also necessary to explore the stability of the resultant material in a variety of simulated oral environments. At the same time, it is necessary to characterize and analyze the structure of the material before and after the reaction.
5. The composition of oral saliva is complex, whether there are some inorganic ions

(e.g. K^+ , Na^+ , Ca^+) and active enzyme species that compete for the charge generated by the pyroelectricity, disturb the trade-off between charge and ROS generation; secondly, the oral saliva contains lysozyme and thiocyanate ion that play a role in oral sterilization, whether cyanide ion can achieve the synergistic bactericidal effect with ROS in this simulated environment, it needs to be further explored.

6. What is the specific role of H_2O_2 in the pro-catalytic effect test? No specific explanation was given in the manuscript. In addition, we know that $\bullet O_2^-$ will be further reduced to generate H_2O_2 . Whether H_2O_2 will be generated during the pyroelectric-catalysis process can be further discussed.

7. In the Fig. 5g-h, it is obvious that in addition to the EPR peak of $\bullet O_2^-$, there are other split peaks generated, indicating the generation of other derived radical intermediates (e.g. $\bullet CH_3$, $\bullet CHO$), which can be further explored. And it is best to supplement the corresponding mechanism study.

Comment 1: In Introduction, more research developments on the application of the host material $BaTiO_3$ in pyroelectric-catalysis should be added.

Response: We thank the reviewer for the suggestion. We have included the application of BTO in pyro-catalysis in the introduction section in page 3 and added refs.25-28.

“As an environmental-friendly lead-free pyroelectric material, $BaTiO_3$ (BTO) has attracted much attention in the field of pyro-catalysis, such as dye degradation by light-induced temperature change and waste heat^{25,26,27}, and wastewater treatment with assistance of metal nanoparticles²⁸.”

Comment 2: In section “Indigo Carmine degradation based on pyro-catalysis”, It says that “17 thermal cycles were required for a 95% degradation of Indigo Carmine at 26-36 °C ($\Delta T = -10$ °C), while only 6 thermal cycles were required for the same degradation of Indigo Carmine at 36-46 °C ($\Delta T = 10$ °C).” This result is not reflected in Supplementary Fig. 4.

Response: Thank you for this comment. The result is from Fig. 2a and Fig 2d rather

than Supplementary Figure 4. We have added the correct figure citations to this result on pages 10-11 of the main text to avoid misunderstanding to the reader:

“17 thermal cycles were required for a 95% degradation of Indigo Carmine at 26-36 °C ($\Delta T = -10$ °C), while only 6 thermal cycles were required for the same degradation of Indigo Carmine at 36-46 °C ($\Delta T = 10$ °C).” has been changed into “13 thermal cycles were required for a 95% degradation of Indigo Carmine at 26-36 °C ($\Delta T = -10$ °C), while only 6 thermal cycles were required for the same degradation of Indigo Carmine at 36-46 °C ($\Delta T = 10$ °C) (Fig. 2a and Fig 2d).”

Comment 3: P11, the fifth line in the third paragraph, “The total charges induced by the pyroelectricity at a rapid heating rate.”, thereinto, how to reflect the charge recombination and release efficiency caused by pyroelectricity? It can be further discussed.

Response: Thank you for this critical comment. We designed a series of experiment to verify that the release of massive free charges leads to a decrease in catalytic efficiency due to the charge recombination. We have performed pyro-catalysis experiments using different catalyst concentrations under the same temperature fluctuation and temperature rise rate. There is no doubt that higher concentration of catalysts will release more charges and more radicals, if the charges do not recombine with each other. However, it was observed that when the catalyst dosage exceeds a certain range, further increase in catalyst concentration has a negative impact on the efficiency of pyro-catalysis. We believe that this is because the simultaneous release of charges in the reaction system is too high, resulting in the recombination of charges with each other instead of reacting with water molecules to form radicals.

The added experimental results were presented in Supplementary Fig.7, and the related further discussion can be found in main text on page 12.

“In order to further discuss the charge recombination and release efficiency on cooling or heating rate, we have additionally performed pyro-catalysis experiment using different catalyst concentrations under the same temperature fluctuation (i.e., $\Delta T = +10$ °C) and temperature rise rate (Supplementary Fig. 7). The pyro-catalytic efficiency

increases and then decreases with the increase of catalyst concentration, indicating that charge release efficiency higher than a center degree will result in the recombination of charges with each other instead of reacting with water molecules to form radicals^{42,43}.”

Supplementary Fig. 7 a-e UV-Vis absorption spectra of Indigo Carmine solutions with respect to BTO concentration. **f** The pseudo-first-order reaction kinetics of different photocatalyst concentration.

Similar results have been obtained in previous studies, where the addition of excess catalyst resulted in a decrease in catalytic performance due to an increase in the charge loading rate. “...But beyond 70 mg, the collision probability between KNN catalysts improves, resulting in the quenching effect between positive charges and negative charges, which leads to a reduction in decomposition ratio.” (*Journal of Cleaner Production*, 2020, 276, 124218.)

Fig. 11. The influence of addition amounts of KNN catalysts on pyroelectric catalysis.

The same conclusion was reached in the piezoelectric catalysis. “Thus, when the additive amount of sodium niobate continues to increase, the quenching between the positive charges and the negative charges causes the decrease of degradation ratio.” (Ceramics International 2019, 45.9, 11703.)

Fig. 7. The sodium niobate nanowires' piezo-catalytic degradation for RhB dye with various initial concentrations under vibration.

Comment 4: In this work, it was relatively scarce in the characterization and analysis of materials. As we know, the human oral environment is complex, in addition to exploring the recyclable activity of the BaTiO₃ nanowires, it is also necessary to explore the stability of the resultant material in a variety of simulated oral environments. At the same time, it is necessary to characterize and analyze the structure of the material

before and after the reaction.

Response: Thank you for this valuable comment. In order to verify BTO nanowires with good stability in the human oral environment, we have carried out the following work:

1) We conducted degradation experiments of indigo carmine using artificial saliva as a solvent for BTO nanowires and BTO gels, respectively. The results show that BTO nanowires still have good cyclic stability in the oral environment simulated by artificial saliva (Supplementary Fig. 11).

2) In addition, we conducted XRD and SEM analysis of BTO nanowires after the catalytic cycling stability characterization in the artificial saliva environment (Supplementary Fig. 12). The results show that the phase and morphology of BTO nanowires did not change before and after the catalytic test in the oral environment simulated by artificial saliva, so it is evident that the BTO nanowires have good cycling performance and stability in the human oral environment. The discussion corresponding to the results is on page 13 in the main text and Supplementary Fig. 11 and 12.

“Due to the complexity of the human oral environment resulting in saliva with many other metal ions as well as enzymes, artificial saliva was employed as a solvent for pyro-catalytic indigo carmine degradation in order to exclude the effect of saliva environmental complexity. The BTO nanowires were subjected to three cycles at a temperature fluctuation of +5 °C. It can be seen that the BTO nanowires exhibit excellent cycling stability in the artificial saliva environment (Supplementary Fig. 11), and their pyro-catalytic performance in this artificial saliva environment is almost the same as that in deionized water (Fig. 3c). In addition, the phase structure and morphology of the BTO nanowires themselves remains stable (Supplementary Fig. 12). This result provides strong support for the application of pyro-catalysis for tooth whitening”

Supplementary Fig. 11 a-c UV-Vis absorption spectra of Indigo Carmine solutions with artificial saliva as solvent using the same BTO nanowires for three cycles. **b** Cyclic stability of BTO nanowires degraded indigo solution.

Artificial saliva (purchased from Shanghai yuanye Bio-Technology Co., Ltd) was used to simulate the real environment of human oral cavity, which is mainly composed of deionized water, NaCl, KCL, Na₂SO₄, NH₄Cl, CaCl₂·2H₂O, NaH₂PO₄·2H₂O, CN₂H₄O, NaF, and this is more than 99% similar to human saliva. After the indigo carmine was dissolved, pyro-catalytic degradation experiments were performed, and the results exhibited in **Supplementary Fig. 11**

Supplementary Fig. 12 a X-ray diffraction pattern of the BTO nanowires before and after pyro-catalysis, and scanning electron microscope image of BTO nanowires **b** before and **c** after pyro-catalysis.

Comment 5: The composition of oral saliva is complex, whether there are some inorganic ions (e.g. K⁺, Na⁺, Ca⁺) and active enzyme species that compete for the charge generated by the pyroelectricity, disturb the trade-off between charge and ROS generation; secondly, the oral saliva contains lysozyme and thiocyanate ion that play a role in oral sterilization, whether cyanide ion can achieve the synergistic bactericidal effect with ROS in this simulated environment, it needs to be further explored.

Response: Thank you for this thoughtful comment. As you said, the real human oral environment is very complex and saliva contains many metal ions as well as enzymes. To ensure that the BTO nanowires can generate free radicals for organic degradation in such a complex environment, we chose artificial saliva as the solvent for Indigo Carmine and performed pyro-catalysis experiments. The results showed that even in the complex environment of saliva, the excellent catalytic performance of the BTO nanowires was not affected by the flammability and showed excellent performance and structural stability. Thus, we conclude that the complexity of the oral environment does not affect the radical production of BTO nanowires. These results is included in the text on page 13 and in Supplementary Information on page 12 and 13.

“Due to the complexity of the human oral environment resulting in saliva with many other metal ions as well as enzymes, artificial saliva was employed as a solvent for pyro-catalytic indigo carmine degradation in order to exclude the effect of saliva environmental complexity. The BTO nanowires were subjected to three cycles at a temperature fluctuation of +5 °C. It can be seen that the BTO nanowires exhibit excellent cycling stability in the artificial saliva environment (Supplementary Fig. 11), and their pyro-catalytic performance in this artificial saliva environment is almost the same as that in deionized water (Fig. 3c). In addition, the phase structure and morphology of the BTO nanowires themselves remains stable (Supplementary Fig. 12). This result provides strong support for the application of pyro-catalysis for tooth

whitening.”

Supplementary Fig. 11 a-c UV-Vis absorption spectra of Indigo Carmine solutions with artificial saliva as solvent using the same BTO nanowires for three cycles. **b** Cyclic stability of BTO nanowires degraded indigo solution

Artificial saliva (purchased from Shanghai yuanye Bio-Technology Co., Ltd) was used to simulate the real environment of human oral cavity, which is mainly composed of deionized water, NaCl, KCL, Na₂SO₄, NH₄Cl, CaCl₂·2H₂O, NaH₂PO₄·2H₂O, CN₂H₄O, NaF, and this is more than 99% similar to human saliva. After the indigo carmine was dissolved, pyro-catalytic degradation experiments were performed.

Supplementary Fig. 12 a X-ray diffraction pattern of the BTO nanowires before and after pyro-catalysis, and scanning electron microscope image of BTO nanowires **b** before and **c** after pyro-catalysis.

In fact, we are working on the use of free radicals for sterilization, as shown in the figure below. We added BTO nanowires to the culture medium and exposed the culture environment to thermal cycles at 20-45°C. From the results of spiral plating, it can be seen that with the help of pyro-catalysis, the bacterial survival rate of the group that added BTO was significantly reduced to 30%, while the bacterial growth of the group that without BTO did not receive a significant effect. This result indicates that the reactive radicals released during pyro-catalysis can be used to achieve bactericidal activity. Based on the present results, it is speculated that the use of free radicals to achieve sterilization with the help of components in the oral environment is highly feasible. This result has been added as Supplementary Fig. 16 and can be found in page 21 in the main text. “The concepts and results strongly highlight the bright prospects of pyroelectric materials used for tooth whitening, or even oral sterilization (Supplementary Fig. 16).”

Supplementary Fig. 16 pyro-catalysis for antibacterial activity in vitro. **a-b** Results of spiral inoculation and **c** cell activity before and after pyro-catalysis.

The pyro-catalytic bacteria sterilization experiment was performed using streptococcus mutans (UA159), which is the culprit of dental plaque and tooth decay. BTO nanowires were first added to EP-tubes for UV sterilization, and then the bacterial solution with an O.D. value of 0.25 at 630 nm was diluted 10,000 times using phosphate buffered saline (PBS). After 400 μl of the diluted solution was added to the sterilized

EP-tubes containing BTO, 20 thermal cycles were performed at 20-45 °C and each cycle took 18 min. Finally, the bacterial solution was transferred to the agar surface using spiral inoculation and their survival was characterized. The results (Supplementary Fig. 16) reveal that the active radicals released by BTO nanowires through pyro-catalysis have a significant bactericidal effect with only 40% survival.

Comment 6: What is the specific role of H₂O₂ in the pro-catalytic effect test? No specific explanation was given in the manuscript. In addition, we know that •O₂⁻ will be further reduced to generate H₂O₂. Whether H₂O₂ will be generated during the pyroelectric-catalysis process can be further discussed.

Response: Thank you for this comment. It is clear that the addition of H₂O₂ significantly accelerated the degradation of indigo carmine based on the experimental results. It has been shown that hydrogen peroxide is involved in the catalytic process as an intermediate product in photocatalysis (see Fig.1 in response letter), and the addition of hydrogen peroxide to the reaction system can induce a Fenton-like effect to enhance the catalytic rate for photocatalysis.

Fig.1 Photocatalytic reaction mechanisms (*Applied Catalysis B: Environmental*, 2020, 272: 118970.)

Analogous to the photocatalysis, we designed experiment to clarify the mechanism of action of H₂O₂ (Supplementary Fig. 3). We monitored the concentration of H₂O₂ in the solution using liquid chromatography during the pyro-catalysis process. The results show that the concentration of H₂O₂ did not change throughout the pyro-catalysis process (Supplementary Fig. 3), so we concluded that there was both consumption and generation of H₂O₂ during the pyro-catalysis process. The added small amount of H₂O₂ combine with the charge released from the pyroelectric material, which then release

free radicals in a Fenton-like reaction. Hence the pyro-catalysis reaction was accompanied by the generation of H_2O_2 , thus achieving a balance between generation and consumption of H_2O_2 .

Additionally, DPD-POD method was further employed to clarify the mechanism of action of H_2O_2 . The related results were shown in Supplementary Fig. 4 (Page 5).

Supplementary Fig. 4 a Typical absorption spectra obtained by testing different concentrations of H_2O_2 using the DPD-POD method. **b** The linear relationship between the H_2O_2 concentration and the peak of the absorption spectrum at 551 nm can be used to calculate the H_2O_2 concentration in the unknown solution. **c** the absorption spectra of H_2O_2 concentrations created by BTO during pyro-catalysis. **d** The concentration of H_2O_2 in the reaction system with and without BTO after a specific number of thermal cycles.

H_2O_2 detection. The concentration of H_2O_2 created during pyro-catalysis process was measured by a DPD-POD method. Briefly, 50mg of BTO was dispersed in 50ml of water, and after a certain number of thermal cycles, samples of the liquid were centrifuged and 1mL of the supernatant was added to a mixture of 3 mL of phosphate buffer (0.5 M, pH = 6), 6 mL of water, 0.05 mL of N,N-diethyl-p-phenylenediamine

sulfate (DPD, 10 mg mL⁻¹) and 0.05 mL of peroxidase (POD, 1 mg mL⁻¹). After 30 s, the H₂O₂ concentration was measured at 551 nm on a UV-vis spectrophotometer.

DPD-POD method was further employed to verify the role of small amount addition of H₂O₂ during pyro-catalysis process^{1,2}. First, the linear relationship between the peak of the absorption spectrum at 551 nm and concentrations of added H₂O₂ indicates the validity of the DPD-POD method, as shown in Supplementary Fig. 4 a and b. It can be seen that the generation of H₂O₂ increase with the number of thermal cycles and about 300 μM of H₂O₂ was produced in the reaction system after 30 hot and cold cycles at a temperature fluctuation of +20 °C, as shown in Supplementary Fig. 4 a and b.

The discussion can be found in main text at page 10, and we also added refs. 39 and 40.

“In contrast, the degradation of Indigo Carmine was minimal when BTO nanowires were used as catalyst or when BTO nanowires were not added (Supplementary Fig. 2). The concentration of H₂O₂ monitored by liquid chromatography shows no change during the pyro-catalysis process (Supplementary Fig. 3), while the DPD-POD experimental results indicate the generation of 300 μM of H₂O₂ in the reaction system after 30 hot and cold cycles at a temperature difference of +20 °C (Supplementary Figs. 4). Anomalous to photo-catalysis, it can be inferred that H₂O₂ is as an intermediate product and remains an equilibrium state between consumption and generation during the pyro-catalysis process. The charges released from the pyro-catalytic BTO nanowires was first combined with the addition of H₂O₂, and subsequently react further into reactive radicals in a Fenton-like reaction^{39,40}. The addition of small amount of H₂O₂ can accelerate the pyro-catalytic process and improve the efficiency. These comparative experiments unambiguously verify that the degradation of Indigo Carmine due to the catalysis effect is strongly associated with the pyroelectricity of the nanowires.”

Comment 7: In the Fig. 5g-h, it is obvious that in addition to the EPR peak of •O₂⁻, there are other split peaks generated, indicating the generation of other derived radical intermediates (e.g. •CH₃, •CHO), which can be further explored. And it is best to

supplement the corresponding mechanism study.

Response: Thank you for this comment. We carefully examined the results of the EPR spectra and there were indeed peaks in the spectra that did not belong to $\bullet\text{O}_2^-$. After comparing them with the literature we confirmed that the additional peaks were from $\bullet\text{CH}_3$. In the detection of $\bullet\text{O}_2^-$, DMSO is used as the solvent, and the $\bullet\text{OH}$ generated in the pyro-catalysis process will react with DMSO, and the $\bullet\text{CH}_3$ generated in the reaction will be captured by DMSO to form $\text{DMSO}\cdot\text{CH}_3$. When the peak of $\text{DMSO}\cdot\text{O}_2^-$ and the peak of $\text{DMSO}\cdot\text{CH}$ appear at the same time, the spectrum in the text will be obtained. *Production of DMPO–OCH3 adduct is most likely initiated by the photocatalytic oxidation of adsorbed water on the titanium dioxide surface, producing reactive hydroxyl radicals, which immediately attack the DMSO solvent. (Journal of Physical Chemistry A, 104,3:557-561)*

Fig. 1. Experimental and simulated EPR spectra (sweep width of 7 mT) measured after 15 min of continuous irradiation ($\lambda > 300$ nm) in $\text{TiO}_2/\text{DMPO}/\text{air}$ suspensions ($c_{\text{DMPO}} = 0.0125 \text{ mol dm}^{-3}$; TiO_2 concentration 3.75 mg ml^{-1}) in following solvents: (a) DMSO, the simulation represents a linear combination of $\bullet\text{DMPO}\cdot\text{O}_2^-$ (S1: $a_{\text{N}} = 1.282 \text{ mT}$, $a_{\text{H}}^{\beta} = 1.035 \text{ mT}$, $a_{\text{H}}^{\alpha} = 0.132 \text{ mT}$; $g = 2.0058$; relative concentration 81%) and $\bullet\text{DMPO}\cdot\text{OCH}_3$ (S2: $a_{\text{N}} = 1.330 \text{ mT}$, $a_{\text{H}}^{\beta} = 0.794 \text{ mT}$, $a_{\text{H}}^{\alpha} = 0.155 \text{ mT}$; $g = 2.0057$; rel. conc. 19%); (b) IPM, the simulation represents a linear combination of $\bullet\text{DMPO}\cdot\text{O}_2^-$ (S1: $a_{\text{N}} = 1.265 \text{ mT}$, $a_{\text{H}}^{\beta} = 0.995 \text{ mT}$, $a_{\text{H}}^{\alpha} = 0.130 \text{ mT}$; $g = 2.0058$; rel. conc. 14%) and $\bullet\text{DMPO}\cdot\text{OR}$ (S2: $a_{\text{N}} = 1.265 \text{ mT}$, $a_{\text{H}}^{\beta} = 0.650 \text{ mT}$, $a_{\text{H}}^{\alpha} = 0.195 \text{ mT}$; $g = 2.0058$; rel. conc. 86%).

We have updated the description of the EPR results on page 13 of the main text and in page 10 of Supplementary information.

“...The additional peaks in Fig. 3i are from the intermediate product DMSO- $\bullet\text{CH}_3$ formed by the reaction of $\bullet\text{OH}$ and DMSO.”

Supplementary Fig. 9 Electron paramagnetic resonance spectra (EPR) of radical **a** $\bullet\text{OH}$ and **b** $\bullet\text{O}_2^-$ created by pyro-catalysis over different cycling times.

In the EPR spectrum of DMPO- $\bullet\text{O}_2^-$, a typical peak belonging to DMPO- $\bullet\text{CH}_3$ was found. This is due to the fact that DMSO is used as an $\bullet\text{OH}$ trapping agent when testing $\bullet\text{O}_2^-$, but at the same time, DMSO is also oxidized by OH to produce CH_3 that cannot be produced by DMPO, and the reaction process can be expressed by the following equations^{5,6}:

REVIEWER COMMENTS

Reviewer #1 (Remarks to the Author):

In the revised manuscript (NCOMMS-21-29906A-Z), titled by "Pyro-catalysis for tooth whitening via oral temperature fluctuation" by Prof. Wang and etc., The issues raised by the reviewer has been answered and responded. Pyroelectricity was used for pyrocatalytic tooth whitening, and it is an interesting and novel work. In my opinion, the revised manuscript can be ccepted by Nature Communications

Reviewer #2 (Remarks to the Author):

After revision, the manuscript has been obviously improved. Before acceptance, some more issues are concerned.

(1) The authors have mentioned some cases that may cause oral temperature fluctuations, however, I am wondering how these cases matching the working conditions required for pyro-catalysis processes. For example, those above-mentioned cases are related to feeding and intaking, usually accompanying with tooth contamination, how would these cases are compatible with pyro-catalysis assisted tooth cleaning processes, which are otherwise associated with falling and degrading the disgusting or even hazardous contaminants from the tooth. Please try to figure out the suitable practical application scenario.

(2) Essentially, the tooth whitening by pyro-catalysis, similar to the photo-catalysis and piezo-catalysis, are achieved via reactive oxidative species (ROSs), such as superoxide $\bullet\text{O}_2^-$ and $\bullet\text{OH}$ radicals, mediated oxidative degradation processes. I am wondering the fundamental differences between the pyro-catalysis and the others, the different generation rate of ROSs? How to understand their big difference in balancing the tooth whitening efficiency and biosafety?

(3) Why BaTiO_3 ? Why BaTiO_3 nanowires? What are the major considerations for choosing a material for tooth whitening, among the available materials? What are the major factors affecting activity of pyro-catalysts? It is strange that the activity is not significantly affected by size of pyro-catalysts, although surface area (A) is an important parameter determining the surface charge.

(4) Thermodynamically, how much energy is needed to generate ROS (equation 3/4) in pyro-catalysis? How to estimate the energy conversion efficiency in pyro-catalysis? Is the temperature change (ΔT) providing sufficient energy to activate oxygen or water molecules?

(5) For oral and intake safety, in addition to the potential harms from ROSs and materials, the pyro-catalysis degradation processes and intermediates shall be also considered.

Reviewer #1:

In the revised manuscript (NCOMMS-21-29906A-Z), titled by "Pyro-catalysis for tooth whitening via oral temperature fluctuation" by Prof. Wang and etc., The issues raised by the reviewer has been answered and responded. Pyroelectricity was used for pyrocatalytic tooth whitening, and it is an interesting and novel work. In my opinion, the revised manuscript can be accepted by Nature Communications.

Response: We are grateful for the referee's positive comments.

Reviewer #2 (Remarks to the Author):

After revision, the manuscript has been obviously improved. Before acceptance, some more issues are concerned.

(1) The authors have mentioned some cases that may cause oral temperature fluctuations, however, I am wondering how these cases matching the working conditions required for pyro-catalysis processes. For example, those above-mentioned cases are related to feeding and intaking, usually accompanying with tooth contamination, how would these cases be compatible with pyro-catalysis assisted tooth cleaning processes, which are otherwise associated with falling and degrading the disgusting or even hazardous contaminants from the tooth. Please try to figure out the suitable practical application scenario.

Response: thanks a lot for this valuable comment. For tooth cleaning, the pyroelectric material could be mixed with hydrogel into dental retainers, patients wearing the retainer can achieve tooth whitening due to oral temperature fluctuations induced by various oral activities such as cold/hot beverages drinking, ice cream eating, and even speaking. However, when patients intake foods, wearing a retainer may cause difficulties for patients to masticate, and even generate hazardous contaminants from the tooth, in this case, the pyro-catalysis is not compatible. Accordingly, the application scenario in Fig. 1a has been revised, the scenario of eating hotpot is replaced with drinking hot tea in the new Fig. 1a.

Additionally, there are many other oral activities that are compatible with pyro-

catalysis assisted tooth cleaning, such as closing and opening the mouth during daily conversations, and breathing during exercise. We have added these appropriate scenarios in the text.

On page 3, “Temperature fluctuations are the most prevalent stimuli in our oral environment, when the mouth is open and close during speaking, drinking, and mouth breathing during exercise. (Fig. 1a).”

On page 5, “These retainers can achieve pyro-catalysis through temperature fluctuations in the mouth induced by daily oral activities (e.g., drinking, breathing, talking, exercising, etc.), without using any other assistant equipment. The generated constant stream of active radicals will attack and degrade the stains on the tooth surface.”

On page 21, “This strategy can be conveniently implemented during our daily oral activities (e.g., drinking, breathing, talking, exercising, etc.) without extra time-consuming and additional equipment.”

The use of retainers provides excellent protection against secondary staining caused by diet, and also prevents degradation products from entering the digestive tract. In addition, a variety of tooth whitening products with peroxide as a whitening agent already exist on the market, we believe that the degradation products of enamel stains will not cause damage to the human body as pyro-catalysis tooth whitening is mechanistically identical to commercial gels (ROS degrades colored macromolecules into small colorless molecules or H₂O and CO₂).

(2) Essentially, the tooth whitening by pyro-catalysis, similar to the photo-catalysis and piezo-catalysis, are achieved via reactive oxidative species (ROSs), such as superoxide $\bullet\text{O}_2^-$ and $\bullet\text{OH}$ radicals, mediated oxidative degradation processes. I am wondering the fundamental differences between the pyro-catalysis and the others, the different generation rate of ROSs? How to understand their big difference in balancing the tooth whitening efficiency and biosafety?

Response: thanks a lot for this comment. As mentioned by the referee, the photo-catalysis, piezo-catalysis and pyro-catalysis are achieved via reactive oxidative species such as $\bullet\text{OH}$ and $\bullet\text{O}_2^-$. However, the mechanisms of these catalytic effects have essential

differences.

1) photo-catalysis

Photo-catalysis uses the semiconductor properties of photocatalysts to convert optical energy into chemical energy. The general mechanism of semiconductor-assisted photo-catalysis is represented in Figure R1, when light energy (photons) hit the semiconductor surface with energy equal to or higher than the bandgap energy. Electrons from the valance band will get excited to the conduction band, leaving holes at the valance band. The holes at the valance band can react with water molecules, generating hydroxyl radicals, which have a strong oxidizing capability that is used to degrade organic matters [Catalysts, 2021, 11].

Fig. R1 General photocatalytic mechanism [Catalysts, 2021, 11].

2) piezo-catalysis

Piezo-catalysis uses mechanical energy as a source and converts it into chemical energy through the piezoelectric effect of piezoelectric nanomaterials. A general working principle or mechanism of the screening charge effect is proposed based on $BaTiO_3$. The $BaTiO_3$ is electrically neutral since the polarization-induced bound charges are equilibrated by the external screening charges with opposite signs at the two polar surfaces (Figure R2a). Under a compressed stress, such as by cavitation bubbles resulting from ultrasound treatment, the temporary charge balance state is destroyed and the polarization will be weakened, leading to the release of extra screening charges from the surface (Figure R2b). These free charges of sufficient energy facilitate redox reactions with diverse substrates (e.g. water molecules, dissolved O_2 , H^+ , and OH^-) in the vicinity of the surface and this stage would last until a new electrical balance is reached (Figure R2c). Once the applied stress is relieved, the

polarization will be restored and more polarization-bound charges will be generated. As a result, space charges from the electrolyte will be adsorbed on the surface again, thereby leaving those charges with opposite polarity (to the adsorbed charges) in the electrolyte to participate in the reactions (Figure R2d). In brief, redox reactions can be accomplished over piezoelectric materials through the polarization-mediated accumulation and release cycles of surface screening charges [Angewandte Chemie, 2022, 134].

Fig. R2 Mechanism of piezo-catalysis [Nature communications, 2020, 11].

3) pyro-catalysis

Pyroelectrics can harvest temperature fluctuation energy to complete redox reactions. The key steps of pyroelectric catalysis are as follows: (i) generation of pyroelectric positive and negative charges; (ii) transformation into reactive species or directly participating in the redox reactions. In the beginning, the pyroelectric crystals stay in surrounding electrolyte to achieve a thermodynamic equilibrium, and the bound polarization charges are fully screened by screening charge carriers. When the temperature varies (ΔT), the equilibrium is disturbed and pyroelectric positive and negative charges located in electronic surface states are generated. The chemical species (OH^- , O_2 , H^+) bearing the opposite charges with the charges from the electrolyte will thus be adsorbed on the surface of the catalysts, and then transformed into ROS, such as hydroxyl radicals ($\bullet\text{OH}$) and superoxide radicals ($\bullet\text{O}_2^-$). The ROS are vital to pyroelectric catalytic oxidation process due to their strong oxidizing power with the capability of mineralizing organics into small molecules (CO_2 , H_2O , etc.) [Nano Energy,

Fig. R3 Mechanism of pyro-catalysis [Nanoscale, 2016, 8].

All the three catalytic processes can release ROS and achieve tooth whitening, however, the shortages of photocatalysis and piezo-catalysis affect their applications in oral cleaning, as listed in Supplementary Table 1. We have added a brief statement on page 2, “However, such photo-catalysis and piezo-catalysis based tooth whitening procedures have been challenged due to either photo-allergic reactions, or the unavailable intrinsic physical stimuli of light irradiation and ultrasonic mechanical vibration (Supplementary Table 1)” and on page 2 of Supplementary information as Supplementary Table 1.

Supplementary Table 1. Comparison of highly effective tooth whitening methods

Tooth Whitening Methods	Materials	Required Physical Field	Damage to Enamel	Shortages
Commercial Gels	Peroxide	None	Erosion	Gingival irritation, mineral loss, and tooth hypersensitivity
Photo-catalysis	Photo-catalyst (TiO ₂)	Blue light	None	Photo-toxic, photo-allergic
Piezo-catalysis	Piezo-catalyst (BaTiO ₃)	Vibration	None	Unavailable intrinsic physical stimuli

On the other side, the pyro-catalysis provides an effective, non-destructive and harmless tooth whitening procedure that requires no additional external energy or specialized equipment. Compared to the commercial gels, pyro-catalysis assistant tooth cleaning is more friendly to the human body, its release of ROS for tooth whitening is slow and continuous, and the low concentration of ROS does not cause damage to tooth enamel, whereas the concentration of ROS released by commercial gels is in much higher level and may harm the tooth enamel. *“Furthermore, the concentration of reactive species generated from BTO nanoparticles is not high enough to damage the enamel, whereas the high concentration of reactive species, and violent nature of their creation from H₂O₂ are such that enamel damage is likely ”*[Nature communications, 2020, 11].

Fig R4 UV-Vis absorption spectra of Indigo Carmine solutions at various vibration time for the poled BTO nanoparticles [Nature communications, 2020, 11].

According to the results of piezoelectric catalytic degradation of Indigo Carmine more than 90% Indigo Carmine was degraded after ultrasonic vibration for 35 min by the poled BTO nano-catalysts. And the pyro-catalytic effect of BTO nanowires exhibited similar performance, more than 90% of Indigo Carmine was degraded in 3 cycles (30 min). It indicates that the quantity of ROS released by pyro-catalysis of BTO nanowires is comparable to that of piezo-catalysis of BTO nanoparticles. As demonstrated by the literature [Nature communications, 2020, 11], the concentration of •OH induced by BTO after vibration is only 1/50 of that released by 3% H₂O₂ of commonly used mouthwash, this is much safer and does not cause damage to the

enamel, which indicates that pyro-catalysis of BTO releasing commensurable ROS is also safe to the human body.

Fig. R5 The EPR spectrum of DMPO-•OH created by Fenton reaction in 3% H₂O₂ (top) and poled BTO after vibrated for 5 min (bottom) [Nature communications, 2020, 11].

The reason why pyro-catalysis does not cause damage to tooth enamel has been added to page 18 of the text. “...stains are removed from the tooth surface, and no damage was caused to the enamel owing to the gentle and continuous release of ROS (Fig. 6b). In contrast, enamel whitened with commercial tooth whitening gels showed significant and irreversible damage to the enamel due to the violent nature of the response caused by the dramatic release of ROS from high peroxide concentrations (Fig. 6c).”

(3) Why BaTiO₃? Why BaTiO₃ nanowires? What are the major considerations for choosing a material for tooth whitening, among the available materials? What are the major factors affecting activity of pyro-catalysts? It is strange that the activity is not significantly affected by size of pyro-catalysts, although surface area (A) is an important parameter determining the surface charge.

Response: thank you for this comment. As teeth whitening materials are so important for oral health, they should be cautiously chosen. There are several factors that should be taken into consideration when we choose materials for pyro-catalysis tooth whitening: 1) The piezoelectric properties of the material, 2) the ability of the material

to generate ROS, and 3) The biosafety of the material.

1) The pyroelectric properties of the material

In this work, tooth whitening is achieved via the pyro-catalytic effect of nanomaterials, so the selected materials must possess favorable pyroelectric properties. BTO is a classical pyroelectric material that is sensitive to temperature changes. *The periodic heating conditions from 2.03 to 13.57 °C are applied by using a semiconductor heater. The positive short-circuit current peak signals can be increased from 1.31 to 12.34 nA corresponding to heating variations. Moreover, the positive peak voltage signals with a 100 MΩ loading resistance can be increased from 0.075 to 0.638 V [Advanced Materials, 2019, 31.].* Temperature changes caused by oral activity have a limited range of temperature fluctuations, and thus BTO is an ideal choice for tooth whitening with an ability to harvest this subtle energy.

Fig. R6 Pyroelectric properties of a single Ag/BTO/Ag sensing device. (a) Measured temperature curves. (b-c) Measured output current and voltage signals under different temperature gradients. [Advanced Materials, 2019, 31.]

2) The ability of the material to generate ROS

For tooth whitening, the selected nanomaterial should release considerable ROS due to temperature fluctuations. ROS (h^+ , $\bullet OH$, and $\bullet O_2^-$) was demonstrated to be

critical in pyroelectric catalysis by BTO. An obvious deactivation of the catalyst was caused by the addition of these radical scavengers, suggesting that h^+ , $\bullet OH$, and $\bullet O_2^-$ are jointly responsible for the degradation of RhB [ACS applied materials & interfaces, 2018, 10].

Fig. R7 (a) Effects of adding different radical scavengers on the pyroelectric catalytic degradation of RhB. (b) The kinetic rate constant for the RhB dye pyroelectric degradation reaction performed with different scavengers. [ACS applied materials & interfaces, 2018, 10]

The ability of BTO nanomaterials to release reactive species through pyro-catalysis was further illustrated by characterizing the variation of ROS concentration with the number of thermal cycles. Here, the high fluorescent 2-hydroxyterephthalic acid is a product of reaction between terephthalic acid and hydroxyl radicals, which is detected to prove the generation of strong oxidative $\bullet OH$. The fluorescence intensity is considered to be proportional to the amount of $\bullet OH$ radicals formed. Thus, the peak located at 425 nm gradually strengthens with the increase of cold-hot cycles, indicating the production of $\bullet OH$ radicals during the pyroelectric catalysis process. Besides, the variation of 2-hydroxyterephthalic acid fluorescence intensity at 425 nm with cold-hot cycles is shown in the inset, which is nearly a line, confirming the continuous and stable production of $\bullet OH$ [Ceramics International, 2020, 46].

Fig. R8 The fluorescence spectrum intensity variation of 2-hydroxyterephthalic acid with respect to cold-hot cycles. The inset is the linear fitting of the fluorescence peak intensity. [Ceramics International, 2020, 46]

3) The biosafety of the material.

Most important of all, the material we choose must be innocuous and friendly to the human body. The application of BTO as a biomaterial has been widely studied for many years. The biocompatibility of BTO was claimed in several studies *in vitro* and *in vivo*. The *in vitro* studies have been done using 4T1 breast cancer cells. The results show that BTO nanoparticles have no effect on cell growth, proliferation and differentiation. And the *in vivo* biocompatibility assessment confirmed that BTO was not translocated to any of the major organs such as liver, kidney, lung, spleen, the heart in the treated mice.

Fig.R9 In vitro piezocatalytic therapy of cancer cells. a) Schematic illustration of cellular level piezocatalytic therapy in vitro. b) Cell viabilities after treatment with T-BTO-Gel at varied concentrations. c) Cell viabilities after exposed to US irradiation ($*p < 0.05$, $**p < 0.01$, $***p < 0.001$). d) Confocal fluorescence images of cancer cells after various treatments, including control group, Gel group, T-BTO-Gel group, Gel + US group and T-BTO-Gel + US group. Scale bar = 40 μm . [Advanced Materials, 2020, 32].

Fig. R10 Histological sections obtained from heart, liver, spleen, lung, and kidney after various treatments, including control group, Gel group, T-BTO-Gel group, Gel + US group and T-BTO-Gel + US group. Scale bar = 200 μm . [Advanced Materials, 2020, 32].

In summary, BTO exhibits promising pyroelectric properties, can release

considerable ROS due to oral temperature fluctuations and has a high level of biosafety, making it an ideal tooth whitening agent.

We selected BTO nanowires for our experiments in order to obtain an improved pyroelectric catalytic performance. According to the results of simulations, nanowires are capable to generate a higher pyroelectric potential and therefore have a better ability to actuate the generation of ROS. BTO nanowires generate a pyroelectric potential as high as 28.2 V under the same conditions with a temperature fluctuation range of 36-56°C, much higher than that generated by nanoparticles. This has been added on page 8 of main text “**And the finite element simulations of pyroelectric potential for different morphological nanomaterials showed that BTO nanowires exhibited outstanding pyroelectric potential compared to other nanostructures due to the spontaneous polarization along the length orientation. (Supplementary Fig. 1).**”, and on page 3 of Supplementary information.

Supplementary Fig. 1 a Temperature fluctuations used in the simulation, and the corresponding pyroelectric potential of BTO nanocrystals in the form of **b** nanoparticle (100 nm×100 nm×100 nm), **c** nanoparticle (200 nm×200 nm×200 nm), and **d** nanowire (100 nm×100 nm×5000 nm).

The pyroelectric potential generated by BTO nanomaterials with different morphologies was simulated by Comsol Multiphysics. The crystal polarization is set along z-axis, and the constitutive relation is described by

$$D = P_s + p(T - T_0) + \varepsilon_0 \varepsilon_r E$$

where T_0 is the initial temperature, D is the electric displacement vector, ε_0 is the permittivity of vacuum, ε_r is the permittivity of BTO, P_s is the spontaneous polarization at T_0 with a value of 0.25 C m^{-2} , p denotes the pyroelectric coefficient which is $210 \text{ } \mu\text{C m}^{-2} \text{ K}^{-1}$, and E is the internal electric field. The other parameters used in this simulation are predefined parameters in Comsol Multiphysics. It can be seen from the results that the pyroelectric potential is distributed along the polar axis. The maximum pyroelectric potential was observed when the largest temperature variation was reached. The comparison of pyroelectric potentials generated by BTO nanomaterials with different morphologies reveals that the pyroelectric potential generated by BTO nanowires is significantly increased, and impressively, the BTO nanowires generated a pyroelectric potential of 28.2 V with a length of $5 \text{ } \mu\text{m}$ in this simulation.

For nanoscale pyro-catalysts, the main factor affecting the pyro-catalytic performance is attributed to the pyroelectric property of the material. The effect of material size on the performance is not significant, which mainly stems from the size effect of ferroelectrics. *The effects of depolarization in small particles have been explained in terms of a randomly oriented surface charge layer that begins to dominate the highly ordered ferroelectric interior as particle size decreases [Chemistry of Materials, 2002, 14]. And a consequence of nanostructuring ferroelectric materials is the appearance of a critical size limit, below which spontaneous polarization cannot be sustained in a ferroelectric material [Journal of Materials Chemistry C, 2013, 1].* The pyro-catalytic effect of BTO comes from the change of ferroelectric polarization induced by temperature fluctuations.

Small BaTiO_3 particles below the critical size d_{crit} of about 30 nm have a cubic phase and are therefore paraelectric and non-pyroelectric. Thus, this group of materials cannot be used as pyro-catalysts. Particles with a higher particle size up to about 100 nm are ferroelectric single crystals with a single ferroelectric domain. The increase of P_r is beneficial for the pyroelectric coefficient and thus increases the pyrocatalytic activity of these particles. Larger particles develop a multidomain structure with a domain size of tens to hundreds of nanometers depending on the

particle size. Individual domains in a multidomain single crystal as well as individual crystallites in a polycrystalline particle should be randomly oriented, resulting in a macroscopic polarization of zero. And a reduction in polarization has a negative effect on the pyroelectric coefficient and thus on the pyrocatalytic activity. [Physical Chemistry Chemical Physics, 2020, 22]

Fig. R11 Schematic representation of the relationships between particlesize, polarization, crystal and domain structure. Typical progressions of polarization versus electric field (P - E) for (A) paraelectric, (B)ferroelectric single-domain, and (C)ferroelectric multidomain crystals. [Physical Chemistry Chemical Physics, 2020, 22]

In general, when the pyro-catalyst size is close to the critical size, the catalytic performance cannot be enhanced due to the decrease in polarization intensity despite the large specific surface area, while the large particles have a larger surface area but the randomly oriented domain structure leads to a decrease in the pyroelectric coefficient, making it unavailable to enhance the catalytic performance. The combination of the above causes leads to a suppressed effect of catalyst size on catalytic performance. BTO nanowires have a size close to the optimum pyroelectric performance of ~ 100 nm, and their growth along the polar axis created strong internal polarization, which enhances the pyroelectric coefficient of materials, thus BTO nanowires can exhibit excellent pyro-catalytic performance.

(4) Thermodynamically, how much energy is needed to generate ROS (equation 3/4) in

pyro-catalysis? How to estimate the energy conversion efficiency in pyro-catalysis? Is the temperature change (ΔT) providing sufficient energy to activate oxygen or water molecules?

Response: thank you for this valuable comment. To achieve pyro-catalysis, the pyroelectric potential generated by pyroelectric materials plays the key role on the generation of ROS. According to the theoretical calculations, *the minimum oxidation potentials of 1.7 and 1.9 V are required for the generation of $\bullet OH$ and $\bullet O_2^-$, respectively [Chemosphere, 2018, 199]*. The pyroelectric potential (Φ_{pyro}), due to temperature changes (ΔT), is governed by:

$$\Phi_{pyro} = \frac{p \cdot l \cdot \Delta T}{\epsilon_0 \cdot \epsilon_r}$$

where p , l and ΔT are the pyroelectric coefficient, nanofiber length and temperature variation, ϵ_0 is the vacuum permittivity (8.854 pF m^{-1}), ϵ_r is the relative permittivity. The reported p and ϵ_r of BTO are $\sim 210 \mu\text{C}\cdot\text{cm}^{-2}\cdot\text{K}^{-1}$ and ~ 100 . For a given ΔT of 5°C , the required length of nanowire is at least $1.6 \mu\text{m}$. The SEM images revealed that the BTO nanowires used in this study are generally $\sim 5 \mu\text{m}$ in length and therefore have the ability to generate $\bullet OH$ and $\bullet O_2^-$. We have added this part of the analysis to the manuscript to refine our work, which can be found in page 12 “**Thermodynamically, the potential for generating $\bullet OH$ and $\bullet O_2^-$ needs to be at least $\sim 1.7 \text{ V}$ and 1.9 V , respectively. The pyroelectric potential (Φ_{pyro}) induced by temperature fluctuations (ΔT) can be governed by Eq. 6.**

$$\Phi_{pyro} = \frac{p \cdot l \cdot \Delta T}{\epsilon_0 \cdot \epsilon_r} \quad (6)$$

where p is the pyroelectric coefficient, l is the length of nanowires, ϵ_0 is the permittivity of vacuum, ϵ_r is the permittivity of BTO. The reported p and ϵ_r of BTO nanowires are $\sim 210 \mu\text{C}\cdot\text{m}^{-2}\cdot\text{K}^{-1}$ and ~ 100 . It can be calculated that the required nanowire length is at least $1.6 \mu\text{m}$ when the temperature fluctuation $\Delta T = 5^\circ\text{C}$. The fabricated BTO nanowires have a length of $\sim 5 \mu\text{m}$, which are capable to realize the pyro-driven ROS generation for degradation of organic dyes.”.

A variety of thermodynamic cycles have been proposed for pyroelectric energy conversion. *While it is possible to envision a Carnot cycle (i.e., two adiabatic ($2 \rightarrow 3$,*

$4 \rightarrow 1$) and two isothermal ($1 \rightarrow 2$, $3 \rightarrow 4$) processes, Fig. 1b), realization of adiabatic processing in a ferroelectric is difficult, making these cycles impractical. Stirling (i.e., two isodisplacement ($2 \rightarrow 3$, $4 \rightarrow 1$) and two isothermal ($1 \rightarrow 2$, $3 \rightarrow 4$) processes, Fig. 1c), Brayton (i.e., two isoelectric ($2 \rightarrow 3$, $4 \rightarrow 1$) and two adiabatic ($1 \rightarrow 2$, $3 \rightarrow 4$) processes, Fig. 1d), and Ericsson (or Olsen) cycles (i.e., two isothermal and isoelectric processes, Fig. 1e), are used in various situations depending on the sample geometry and heat source. The Olsen cycle has been most widely employed and has been demonstrated to produce some of the highest PEC efficiencies, defined as:

$$\eta = \frac{\oint EdP}{\int_{T_L}^{T_H} C(T)dT + Q_{ECE}}$$

where $C(T)$ is the heat capacity, T_H (T_L) is the temperature for the heat source (heat sink), $\oint EdP$ is the net electrical work done (W_E), and Q_{ECE} is the electrocaloric work.

Fig. R12 a Schematic illustrating the pyroelectric effect as a change in polarization with temperature. Polarization vs. electric-field pathways for b Carnot, c Stirling, d Brayton, and e Ericsson (Olsen) cycles. [NPG Asia Materials, 2019, 11]

Theoretically, the pyroelectric effect has a high energy conversion efficiency when pyroelectric materials are used as energy harvesting devices. A pyroelectric heat engine

working in a thermal cycle can, in theory, generate usable electrical power reaching 84-92% Carnot efficiency [Energy & Environmental Science, 2018, 11]. And when used as a nano-catalyst, it is commonly difficult to calculate its energy conversion efficiency. Based on the experimental results of hydrogen evolution by pyro-catalysis and comparing the performance of pyro-catalysis and photocatalysis, the energy conversion efficiency of pyro-catalysts can be roughly derived. In theory, energy conversion efficiency (40%–45%) of pyroelectric materials should be higher than that of photovoltaic materials (<20%) [Nano Energy, 2020, 78].

(5) For oral and intake safety, in addition to the potential harms from ROSs and materials, the pyro-catalysis degradation processes and intermediates shall be also considered.

Response: thank you for this comment. The tooth whitening through ROSs has been used in the commercial tooth whitening gel for many years, the degradation processes and intermediates have been confirmed to be harmless to human body. In fact, the composition of the tooth stains is so complicated that have not been clarified [J Evid Based Dent Pract, 14 70-6 (2014)], wherein the deposition of the chromogen is one of the culprits that cause tooth stains. For example, Indigo Carmine, the commonly used chromogen in food additives, can be degraded by ROS through commercial tooth whitening gel. The intermediate of the degradation process of Indigo Carmine via ROS is mainly isatin-5-sulfonic acid (See Figure R13) [Journal of Photochemistry and Photobiology A: Chemistry, 2021, 405.], which has been well studied and confirmed to be biosafe. As demonstrated in the literature [Water, Air, & Soil Pollution, 2022, 233.]: *In order to determine which of the components of the laundry wastewater were responsible for the toxic effects, the D. magna test was used to individually analyze the toxicity of AB74, NaCl, and Biolite BSN. Furthermore, since laccases can oxidize AB74 to isatin 5-sulfonic acid and probably to anthranilic acid, the toxicity of these two compounds was also evaluated. AB74, isatin 5-sulfonic acid, anthranilic acid, and NaCl were shown to be nontoxic at the range of concentrations evaluated (0–100 mg/L for AB74, isatin 5-sulfonic acid and anthranilic acid; 0-3 g/L for NaCl).*

Thus, we may conclude that the pyro-catalytic tooth whitening via ROS will not produce harmful intermediates.

Fig. R13 Negative mode ESI-MS spectra of IC. [Journal of Photochemistry and Photobiology A: Chemistry, 2021, 405.]

To provide additional insight into the degradation pathway of IC, negative mode ESI-MS was employed as a characterization technique. Initial analysis of pure IC was performed for comparative purposes showing ions at 210, 421, and 443 m/z associated with $[IC-2Na]^{2-}$, $[IC+H-2Na]^{-}$ and $[IC-Na]^{-}$, respectively. The colorless solution obtained after 60 min of irradiation established isatin-5-sulfonic acid (226 m/z) as the major IC degradation product. Tandem MS experiments confirmed the identity of this compound. Also, fragments at 216 and 244 m/z were identified as 4-amino-3-carboxybenzenesulfonate and 4-amino-3-(carboxycarbonyl) benzenesulfonate, respectively. No IC characteristic ions remained after the irradiation cycle indicating total dye-degradation. And it is also confirmed that the isatin-5-sulfonic acid (226 m/z) is the primary degradation product. A similar MS profile was obtained for the solution where oxalic acid was employed as a sacrificial electron donor for the photocatalytic process (with just 5 min of irradiation), being the isatin-5-sulfonic acid (226 m/z) the primary degradation product as well. [Journal of Photochemistry and Photobiology A: Chemistry, 2021, 405.]

Table 1 Acute toxicity evaluation with *Daphnia magna*

	LC ₅₀ (%)	95% Confidence interval		Toxicity level
		Lower bound	Upper bound	
Simulated denim-laundry wastewater	29.7 a	20.0	32.4	Toxic
AB74 (100 mg/L)	NT	NT	NT	Nontoxic
Isatin 5-sulfonic acid (100 mg/L)	NT	NT	NT	Nontoxic
Anthranilic acid (100 mg/L)	NT	NT	NT	Nontoxic
NaCl (3 g/L)	NT	NT	NT	Nontoxic
Biolite BSN (5 g/L)	24.3 a	13.0	39.4	Very toxic
UASB effluent	63.3 b	53.1	77.3	Moderately toxic

LC₅₀ lethal concentration 50. NT non-toxic under the proved conditions. Values with different lowercase letters within the same column indicate significant differences ($p < 0.05$)

Fig. R14 Acute toxicity evaluation of isatin 5-sulfonic acid. [Water, Air, & Soil Pollution, 2022, 233.]

REVIEWERS' COMMENTS

Reviewer #2 (Remarks to the Author):

The work demonstrates the potential of pyro-catalysis for tooth whitening, I am satisfied with the responses and revisions by the authors, and the revised version can be accepted. Anyway, I am still concerning about the feasibility and safety of the pyrocatalytic technique for practical applications, which would require further considerations and investigations.

(1) In Fig 1a, the eating scenario is replaced by drinking in revised version, but this change obviously cannot prevent the pyro-catalysis induced degradation products from entering the digestive tract.

On the other hand, the market-available peroxide-dominant tooth whitening products are normally not used during feeding or intaking, and, the degradation product could be immediately spat out by the customers, avoiding the possible damage to human body.

Based on the above comments, it would better find suitable practical application scenario where pyro-catalysis would work separately from regular daily activities.

(2) Advanced oxidation processes involving the oxidation reactions of organic compounds by oxidative radical species, such as $\bullet\text{O}_2^-$ and $\bullet\text{OH}$, often produce unpredictable intermediates that may be harmful, even the original organic compound is totally not toxic. For example, a similar concern is especially serious for drinking water disinfection treatment, where the production of toxic disinfection byproducts is usually one of the major concerns about drinking water safety.

I hope the authors would make some frank comments on the above concerns in the final version.

REVIEWERS' COMMENTS

Reviewer #2 (Remarks to the Author):

The work demonstrates the potential of pyro-catalysis for tooth whitening, I am satisfied with the responses and revisions by the authors, and the revised version can be accepted. Anyway, I am still concerning about the feasibility and safety of the pyrocatalytic technique for practical applications, which would require further considerations and investigations.

(1) In Fig 1a, the eating scenario is replaced by drinking in revised version, but this change obviously cannot prevent the pyro-catalysis induced degradation products from entering the digestive tract.

On the other hand, the market-available peroxide-dominant tooth whitening products are normally not used during feeding or intaking, and, the degradation product could be immediately spat out by the customers, avoiding the possible damage to human body. Based on the above comments, it would better find suitable practical application scenario where pyro-catalysis would work separately from regular daily activities.

Response: thank you for this suggestion. At beginning, we also concern the risk of degradation products and ferroelectric particles entering the digestive tract during the tooth whitening process. After consulting with dentists, our pyro-catalysis tooth whitening strategy is feasible because the stained teeth were covered by the whitening retainer, in turn degradation products can be prevented from entering the digestive tract. Moreover, the degradation products can be released by rinsing mouth at the end of whitening, which reduces the risk of degradation products to human body.

(2) Advanced oxidation processes involving the oxidation reactions of organic compounds by oxidative radical species, such as $\bullet\text{O}_2^-$ and $\bullet\text{OH}$, often produce unpredictable intermediates that may be harmful, even the original organic compound is totally not toxic. For example, a similar concern is especially serious for drinking water disinfection treatment, where the production of toxic disinfection byproducts is usually one of the major concerns about drinking water safety.

Response: thank you for this valuable comment. The structure of organic matter is very complex, so there are multiple reaction pathways in the mineralization to H_2O and CO_2 , which will probably lead to some unpredictable intermediates. However, the amount of unpredictable intermediates is very limited, which were usually extremely scarce or even undetectable. While pyro-catalytic tooth whitening is an efficient and time-saving process, we believe that the production of unpredictable intermediates will be more scarce in this short time period. The dentists told us the possibility of harmful unpredictable intermediates should be much weaker than the market-available peroxide-dominant tooth whitening products. For example, Indigo Carmine (a commonly used chromogen in food additives) is an essential element to form tooth stains, can be degraded by ROS, the degradation pathways of Indigo Carmine have been reported in detail [*Materials Science in Semiconductor Processing* 2020, 105.]. *Indigo Carmine degradation was analyzed by analytical techniques,*

UPLC-PDA and HR-QTOF ESI/MS that helps to identify the degradation products, organic reactions (Hydroxylation, oxidation, methylation, decarboxylation, and desulfonation), and four pathways of Indigo Carmine in water. All the reaction intermediates that can be detected are NOT harmful to human body.

Fig. R1 Proposed degradation pathways of IC in water by visible light by Ni-BaMo₃O₁₀ photocatalyst.

[Materials Science in Semiconductor Processing 2020, 105.]